# Sequence co-evolution gives 3D contacts and structures of protein complexes

**Thomas A Hopf[1,2†], Charlotta P I Schärfe[1,3,4†], João P G L M Rodrigues[5†], Anna G Green[1], Oliver Kohlbacher[3,4], Chris Sander[6*], Alexandre M J J Bonvin[5*], Debora S Marks[1*]**

[1]Department of Systems Biology, Harvard University, Boston, United States; [2]Bioinformatics and Computational Biology, Department of Informatics, Technische Universität München, Garching, Germany; [3]Applied Bioinformatics, Quantitative Biology Center, University of Tübingen, Tübingen, Germany; [4]Department of Computer Science, University of Tübingen, Tübingen, Germany; [5]Computational Structural Biology Group, Bijvoet Center for Biomolecular Research, Utrecht University, Utrecht, Netherlands; [6]Computational Biology Center, Memorial Sloan Kettering Cancer Center, New York, United States

**Abstract** Protein–protein interactions are fundamental to many biological processes. Experimental screens have identified tens of thousands of interactions, and structural biology has provided detailed functional insight for select 3D protein complexes. An alternative rich source of information about protein interactions is the evolutionary sequence record. Building on earlier work, we show that analysis of correlated evolutionary sequence changes across proteins identifies residues that are close in space with sufficient accuracy to determine the three-dimensional structure of the protein complexes. We evaluate prediction performance in blinded tests on 76 complexes of known 3D structure, predict protein–protein contacts in 32 complexes of unknown structure, and demonstrate how evolutionary couplings can be used to distinguish between interacting and non-interacting protein pairs in a large complex. With the current growth of sequences, we expect that the method can be generalized to genome-wide elucidation of protein–protein interaction networks and used for interaction predictions at residue resolution.

**\*For correspondence:** sander@cbio.mskcc.org (CS); a.m.j.j.bonvin@uu.nl (AMJJB); debbie@hms.harvard.edu (DSM)

†These authors contributed equally to this work

**Competing interests:** The authors declare that no competing interests exist.

**Reviewing editor**: John Kuriyan, Howard Hughes Medical Institute, University of California, Berkeley, United States

## Introduction

A large part of biological research is concerned with the identity, dynamics, and specificity of protein interactions. There have been impressive advances in the three-dimensional (3D) structure determination of protein complexes which has been significantly extended by homology-inferred 3D models (*Mosca et al., 2012*; *Webb et al., 2014*) (*Hart et al., 2006*; *Zhang et al., 2012*). However, there is still little, or no, 3D information for ~80% of the currently known protein interactions in bacteria, yeast, or human, amounting to at least ~30,000/~6000 incompletely characterized interactions in human and *Escherichia coli*, respectively (*Mosca et al., 2012*; *Rajagopala et al., 2014*). With the rapid rise in our knowledge of genetic variation at the sequence level, there is an increased interest in linking sequence changes to changes in molecular interactions, but current experimental methods cannot match the increase in the demand for residue-level information of these interactions. One way to address the knowledge gap of protein interactions has been the use of hybrid, computational–experimental approaches that typically combine 3D structural information at varying resolutions, homology models, and other methods (*de Juan et al., 2013*), with force field-based approaches such as RosettaDock, residue cross-linking, and data-driven approaches that incorporate various sources of biological information (*Kortemme and Baker, 2002*; *Dominguez et al., 2003*; *Kortemme and Baker, 2004*;

**eLife digest** DNA is often referred to as the 'blueprint of life', as this molecule contains the instructions that are required to build a living organism from a single cell. But these instructions largely play out through the proteins that DNA encodes; and most proteins do not work alone. Instead they come together in different combinations, or complexes, and a single protein may participate in many complexes with different activities.

Proteins are so small that it is difficult to get clear information about what they look like. Visualizing protein complexes is even harder. Most protein–protein interactions remain poorly understood, even in the best-studied organisms such as humans, yeast, and bacteria.

Proteins are made from smaller molecules, called amino acids, strung together one after the other. The order in which different amino acids are arranged in a protein determines the protein's shape and ultimately its function. Like DNA, protein sequences can change over time. Sometimes, the sequence of one protein changes in a way that prevents it binding to another protein. If these two proteins must work together for an organism to survive, the second protein will often develop a compensating change that allows the protein–protein complex to reform.

Identifying pairs of changes in the sequences of pairs of proteins suggests that the two proteins interact and gives some information about how the proteins fit together. Different species can have copies of the same proteins that have slightly different sequences. Since the DNA sequences from many different organisms are already known, there are now many opportunities to find sites in pairs of proteins that have evolved together, or co-evolved, over time.

To find sites that seem to have co-evolved, Hopf et al. used a computer program based on an approach from statistical physics to look at pairs of proteins that were already known to form complexes. Co-evolving sites were found in over 300 pairs of proteins; including 76 where the structure of the complex was already known. When sites that were predicted to be co-evolving were then mapped to these known complex structures, the co-evolving sites were remarkably close to the true protein–protein contacts. This indicates that the information from the co-evolved sequences is sufficient to show how two proteins fit together.

Hopf et al. then turned their attention to 82 pairs of proteins that were thought to interact, but where a structure was unavailable. For 32 of these pairs, structures of the entire complex could be predicted, showing how the two proteins might interact. Furthermore, when other researchers subsequently worked out the structure of one of these complexes, the prediction was a good match to the solved complex structure.

The machinery of life is largely made up of proteins, which must interact in ever-changing but precise ways. The new methods developed by Hopf et al. provide a new way to discover and investigate the details of these interactions.

*Kortemme et al., 2004*; *Svensson et al., 2004*; *Chaudhury et al., 2011*; *Schneidman-Duhovny et al., 2012*; *Velazquez-Muriel et al., 2012*; *Karaca and Bonvin, 2013*; *Rodrigues et al., 2013*; *Webb et al., 2014*). However, most of these approaches depend on the availability of prior knowledge and many biologically relevant systems remain out of reach, as additional experimental information is sparse (e.g., membrane proteins, transient interactions, and large complexes). One promising computational approach is to use evolutionary analysis of amino acid co-variation to identify close residue contacts across protein interactions, which was first used 20 years ago (*Gobel et al., 1994*; *Pazos and Valencia, 2001*), and subsequently used also to identify protein interactions (*Pazos et al., 1997*; *Pazos and Valencia, 2002*). Others have used some evolutionary information to improve a machine learning approach to developing docking potentials (*Faure et al., 2012*; *Andreani et al., 2013*; *Andreani and Guerois, 2014*). These previous approaches relied on a local model of co-evolution that is less likely to disentangle indirect and therefore incorrect correlations from the direct co-evolution, as has been described in work on residue–residue interactions in single proteins (*Marks et al., 2012*). More recently, reports using a global model have been successful in identifying residue interactions from evolutionary co-variation, for instance between histidine kinases and response regulators (*Burger & van Nimwegen, 2008*; *Skerker et al., 2008*; *Weigt et al., 2009*), and this approach has only recently been generalized and used to predict contacts between proteins in complexes of unknown structure, in an independent effort

parallel to this work (*Ovchinnikov et al., 2014*). In principle, just a small number of key residue–residue contacts across a protein interface would allow computation of 3D models and provide a powerful, orthogonal approach to experiments.

Since the recent demonstration of the use of evolutionary couplings (ECs) between residues to determine the 3D structure of individual proteins (*Marks et al., 2011*; *Morcos et al., 2011*; *Aurell and Ekeberg, 2012*; *Jones et al., 2012*; *Kamisetty et al., 2013*), including integral membrane proteins (*Hopf et al., 2012*; *Nugent and Jones, 2012*), we reason that an evolutionary statistical approach such as EVcouplings (*Marks et al., 2011*) could be used to determine co-evolved residues *between* proteins. To assess this hypothesis, we built an evaluation set based on all known binary protein interactions in *E. coli* that have 3D structures of the complex as recently summarized (*Rajagopala et al., 2014*). We develop a score for every predicted inter-protein residue pair based on the overall inter-protein EC score distributions resulting in accurate predictions for the majority of top ranked *inter*-protein EC pairs (inter-ECs) and sufficient to calculate accurate 3D models of the complexes in the docked subset (*Figure 1A*). This approach was then used to predict evolutionary couplings for 32 complexes of unknown 3D structures that have a sufficient number of sequences. Predictions include

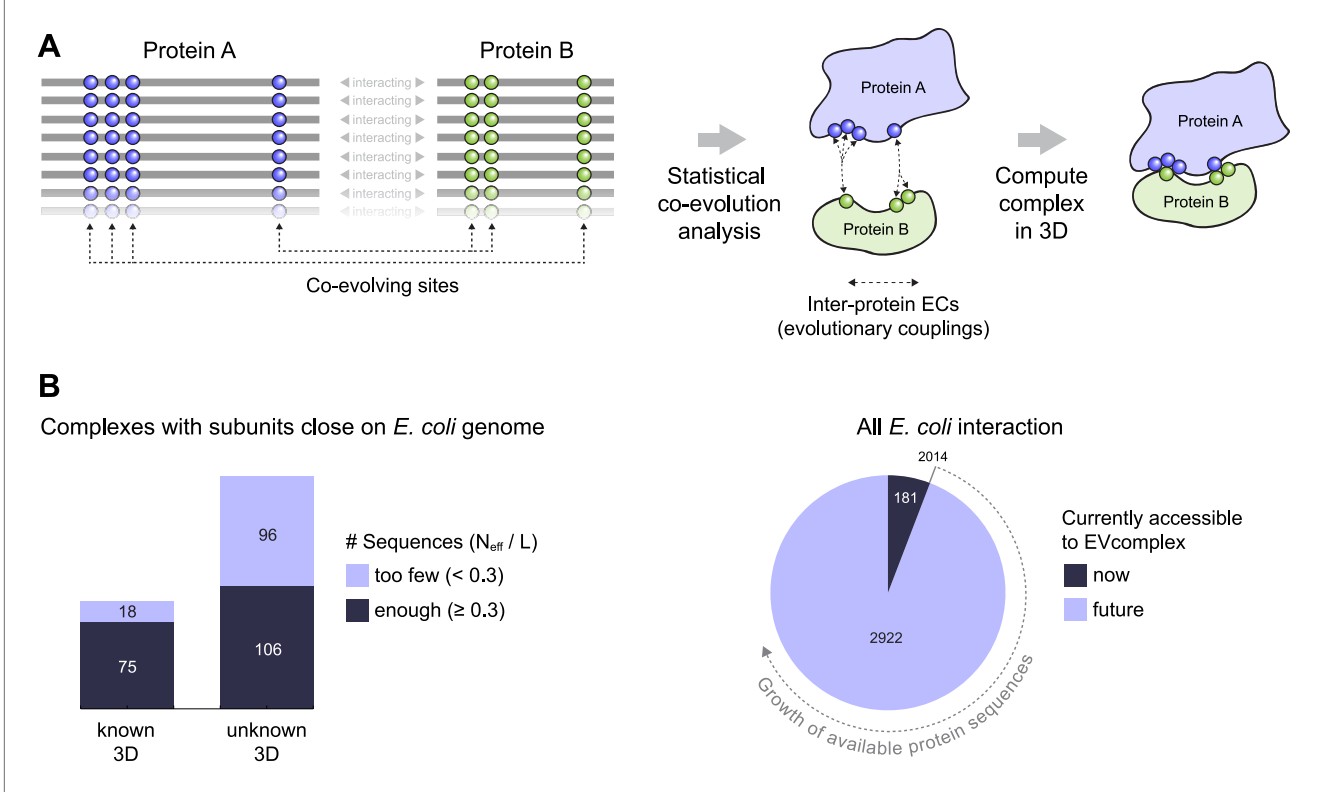

**Figure 1**. Co-evolution of residues across protein complexes from the evolutionary sequence record. (**A**) Evolutionary pressure to maintain protein–protein interactions leads to the co-evolution of residues between interacting proteins in a complex. By analyzing patterns of amino acid co-variation in an alignment of putatively interacting homologous proteins (left), evolutionary couplings between co-evolving inter-protein residue pairs can be identified (middle). By defining distance restraints on these pairs, the 3D structure of the protein complex can be inferred using docking software (right). (**B**) Distribution of *E. coli* protein complexes of known and unknown 3D structure where both subunits are close on the bacterial genome (left), allowing sequence pair matching by genomic distance. For a subset of these complexes, sufficient sequence information is available for evolutionary couplings analysis (dark blue bars). As more genomic information is created through on-going sequencing efforts, larger fractions of the *E. coli* interactome become accessible for EVcomplex (right). A detailed version of the workflow used to calculate all *E. coli* complexes currently for which there is currently enough sequence information is shown in *Figure1—figure supplement 1*.

The following figure supplement is available for figure 1:

**Figure supplement 1**. Details of the EVcomplex Pipeline.

the structurally unsolved interactions between the a-, b-, and c-subunits of ATP synthase, which are supported by previously published experimental results.

## Results

We first investigated whether co-evolving residues between proteins are close in three dimensions by assessing blinded predictions of residue co-evolution against experimentally determined 3D complex structures. We follow this evaluation by then predicting co-evolved residue pairs of interacting proteins that have no known complex structure.

### Extension of the evolutionary couplings method to protein complexes

To compute co-evolution across proteins, individual protein sequences must be paired up with each other that are presumed to interact, or being tested to see if they interact. Without this condition, proteins could be paired together that do not in fact interact with each other and therefore detection of co-evolution would be compromised. Given that the evolutionary couplings method depends on large numbers of diverse sequences (*Hopf et al., 2012*), some assumption must be made about which proteins interact with each other in homologous sequences in other species. Since it is challenging to know a priori whether particular interactions are conserved across many millions of years in thousands of different organisms, we use proximity of the two interacting partners on the genome as a proxy for this, with the goal of reducing incorrect pairings.

To assemble the broadest possible data sets to test the approach and make predictions, we take all known interacting proteins assembled in a published data set that contains ~3500 high-confidence protein interactions in *E. coli* (*Rajagopala et al., 2014*). After removing redundancy and requiring close genome distance between the pairs of proteins this results in 326 interactions, see 'Materials and methods' (*Figure 1B*, *Figure 1—figure supplement 1*, *Supplementary file 1 and 2*),

The paired sequences are concatenated and statistical co-evolution analysis is performed using EVcouplings (*Marks et al., 2011*; *Morcos et al., 2011*; *Aurell and Ekeberg, 2012*), that applies a pseudolikelihood maximization (PLM) approximation to determine the interaction parameters in the underlying maximum entropy probability model (*Balakrishnan et al., 2011*; *Ekeberg et al., 2013*; *Kamisetty et al., 2013*), simultaneously generating both intra- and inter-EC scores for all pairs of residues within and across the protein pairs (*Figure 1A*). Evolutionary coupling calculations in previous work have indicated that this global probability model approach requires a minimum number of sequences in the alignment with at least 1 non-redundant sequence per residue (*Marks et al., 2011*; *Morcos et al., 2011*; *Hopf et al., 2012*; *Jones et al., 2012*; *Kamisetty et al., 2013*). Our current approach allows complexes with fewer available sequences to be assessed (minimum at 0.3 non-redundant sequences per residue) by using a new quality assessment score to assess the likelihood of the predicted contacts to be correct. The EVcomplex score is based on the knowledge that most pairs of residues are not coupled and true pair couplings are outliers in the high-scoring tail of the distribution (See 'Materials and methods', *Figure 2A,B*, *Figure 2—figure supplement 1 and 2*). The score can intuitively be understood as the distance from the noisy background of non-significant pair scores, normalized by the number of non-redundant sequences and the length of the protein ('Materials and methods', *equations 1* and *2*). If the number of sequences per residue is not controlled for, there is a large bias in the results, overestimating performance with low numbers of sequences (*Figure 2B,C*). The precise functional form of the correction for low numbers of sequences was chosen non-blindly after observing the dependencies in the test set.

### Blinded prediction of known complexes

#### Evolutionary covariation reveals inter-protein contacts

Of the 329 interactions identified that are close on the *E. coli* genome, 76 have a sufficient number of alignable homologous sequences and known 3D structures either in *E. coli* or in other species. This set was used to test the inter-protein evolutionary coupling predictions (*Supplementary file 1*). The relationship between the EVcomplex score and the precision of the corresponding inter-protein ECs suggests that on average 74% (69%) of the predicted pairs with EVcomplex score greater than 0.8 will be accurate to within 10 Å (8 Å) of an experimental structure of the complex (*Figure 2C*). Most complexes have at least one inter-protein predicted contact above the selected score threshold of 0.8 (53/76 complexes). Three complexes have more than 20 predicted inter-protein residue contacts which are over 80% accurate, namely the histidine kinase and response regulator system (78 residue pairs), t-RNA

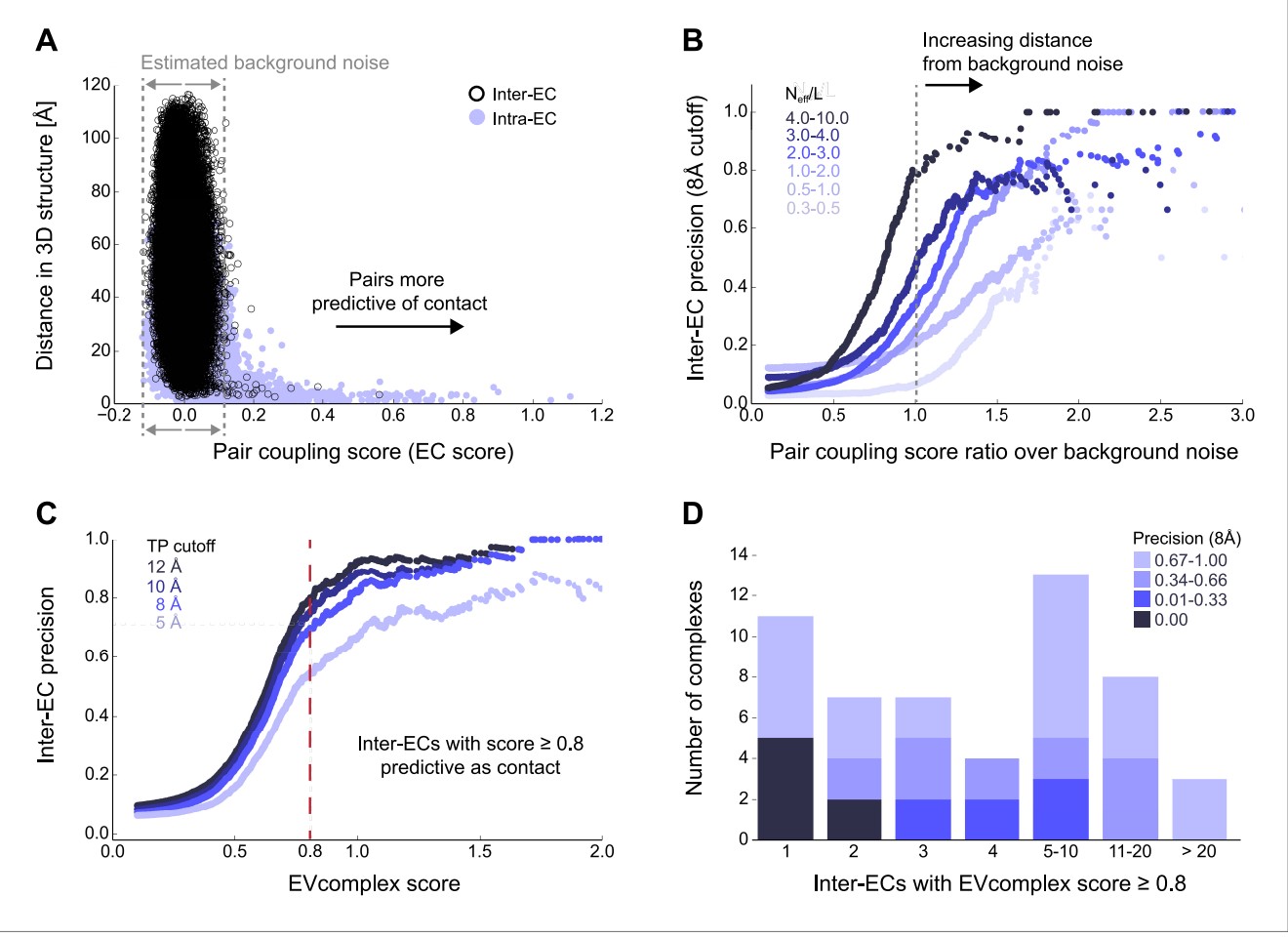

**Figure 2**. Evolutionary couplings capture interacting residues in protein complexes. (**A**) Inter- and Intra-EC pairs with high coupling scores largely correspond to proximal pairs in 3D, but only if they lie above the background level of the coupling score distribution. To estimate this background noise a symmetric range around 0 is considered with the width being defined by the minimum inter-EC score. For the protein complexes in the evaluation set, this distribution is compared to the distance in the known 3D structure of the complex that is shown here for the methionine transporter complex, MetNI. (Plots for all complexes in the evaluation set are shown in *Figure 2—figure supplement 1 and 2*.) (**B**) A larger distance from the background noise (ratio of EC score over background noise line) gives more accurate contacts. Additionally, the higher the number of sequences in the alignment the more reliable the inferred coupling pairs are which then reduces the required distance from noise (different shades of blue). Residue pairs with an 8 Å minimum atom distance between the residues are defined as true positive contacts, and precision = TP/(TP + FP). The plot is limited to range (0,3) which excludes the histidine kinase—response regulator complex (HK–RR)—a single outlier with extremely high number of sequences. (**C**) To allow the comparison across protein complexes and to estimate the average inter-EC precision for a given score threshold independent of sequence numbers, the raw couplings score is normalized for the number of sequences in the alignment, resulting in the EVcomplex score. In this work, inter-ECs with an EVcomplex score ≥0.8 are used. Note: the shown plot is cut off at a score of 2 in order to zoom in on the phase change region and the high sequence coverage outlier HK-RR is excluded. (**D**) For complexes in the benchmark set, inter-EC pairs with EVcomplex score ≥0.8 give predictions of interacting residue pairs between the complex subunits to varying accuracy (8 Å TP distance cutoff). All predicted interacting residues for complexes in the benchmark set that had at least one inter-EC above 0.8 are shown as contact maps in *Figure 2—figure supplement 3–8*.

The following figure supplements are available for figure 2:

**Figure supplement 1**. Distribution and accuracy of raw EC scores for all complexes in evaluation set.

**Figure supplement 2**. Distribution and accuracy of raw EC scores for all complexes in evaluation set (2).

**Figure supplement 3**. Contact maps of all complexes with solved 3D structure with inter-ECs above EVcomplex score of 0.8.

**Figure supplement 4**. Contact maps of all complexes with solved 3D structure with inter-ECs above EVcomplex score of 0.8 (2).

*Figure 2. Continued*

**Figure supplement 5**. Contact maps of all complexes with solved 3D structure with inter-ECs above EVcomplex score of 0.8 (3).

**Figure supplement 6**. Contact maps of all complexes with solved 3D structure with inter-ECs above EVcomplex score of 0.8 (4).

**Figure supplement 7**. Contact maps of all complexes with solved 3D structure with inter-ECs above EVcomplex score of 0.8 (5).

**Figure supplement 8**. Contact maps of all complexes with solved 3D structure with inter-ECs above EVcomplex score of 0.8 (6).

synthetase (32 residue pairs), and the vitamin B importer complex (21 residue pairs), with precision over 80% (complex numbers 330, 019, 130 respectively, *Figure 2D*, *Figure 2—figure supplements 3–8*, *Supplementary file 1*).

We suggest that users of EVcomplex consider predicted contacts that lie below the threshold of 0.8 in the context of other biological knowledge, where available, or in comparison to other higher scoring contacts for the same complex. In this way additional true positive inter-residue contacts can be distinguished from false positives. For instance, the ethanolamine ammonia-lyase complex (complex 065) has only 3 predicted inter-protein residue pairs above the score threshold, but in fact has 5 additional correct pairs with EVcomplex scores slightly below the threshold of 0.8 which cluster with the 3 high-scoring contacts on the monomers, indicating that they are also correct.

Some of the high confidence inter-protein ECs in the test set are not close in 3D space when compared to their known 3D structures. These false positives may be a result of assumptions in the method that are not always correct. This includes (1) the assumption that the interaction between paired proteins is conserved across species and across paralogs, and (2) that truly co-evolved residues across proteins are indeed always close in 3D, which is not always the case. In addition, the complexes may also exist in alternative conformations that have not necessarily all been captured yet by crystal or NMR structures, for instance in the case of the large conformational changes of the BtuCDF complex (*Hvorup et al., 2007*).

## Docking is accurate with few pairs of predicted contacts

To test whether the computed inter-protein ECs are sufficient for obtaining accurate 3D structures of the whole complex, we selected 15 diverse examples (with 5 or more inter-protein residue contacts) for docking (*Table 1*, *Figure 3*, *Figure 3—figure supplement 1*, *Supplementary file 3*) with HADDOCK (*Dominguez et al., 2003*; *de Vries et al., 2007*). The docking procedure is fast and generates 100 3D models of each complex using all residue pairs with EVcomplex scores above the selection threshold. We additionally dock negative controls to assess the amount of information added to the docking protocol by evolutionary couplings (500 models per run, no constraints other than center of mass, see 'Materials and methods'). The best models for all 15 complexes docked with evolutionary couplings have interface RMSDs under 6 Å, 12/15 have the best scoring model under 4 Å and the top ranked models for 11/15 are under 5 Å backbone interface RMSD compared to a crystal or NMR structure interface. Over 70% of the generated models are close to the experimental structures of the complexes (<4 Å backbone iRMSD), compared to less than 0.5% in the controls (and these were not high–ranked) (*Figure 3—figure supplement 1*, *Supplementary file 3*, 'Supplementary data'). Not surprisingly complexes that have the largest numbers of true positive predicted contacts perform the best when docking. For example, the ribosomal proteins RS3 and RS14 have 11 true positive inter-protein ECs and result in a top ranked model only 1.1 Å iRMSD from the reference structure. More surprisingly, other complexes with a lower proportion of true positive inter-protein contacts, such as Ubiquinol oxidase (6 out of 11) or the epsilon and gamma subunits of ATP synthase (8 out of 15) also produced accurate predicted complexes, with an iRMSD of 1.8 and 1.4 Å respectively. The docking experiments therefore demonstrate that inter-protein ECs, even in the presence of incorrect predictions, can be sufficient to give accurate 3D models of protein complexes, but more work will be needed to quantify the likelihood of successful docking from the predicted contacts.

## Conserved residue networks provide evidence of functional constraints

The top 10 inter-EC pairs between MetI and MetN are accurate to within 8 Å in the MetNI complex (PDB: 3tui [*Johnson et al., 2012*]), resulting in an average 1.4 Å iRMSD from the crystal structure for

**Table 1.** EVcomplex predictions and docking results for 15 protein complexes

| Complex name | Subunits | EVcomplex contacts | | | Docking quality (iRMSD) | |
|---|---|---|---|---|---|---|
| | | Seqs† | ECs‡ | TP rate§ | Top ranked model# | Best model¶ |
| Carbamoyl-phosphate synthase | CarB:CarA | 2.3 | 17 | 0.88 | 1.9 | 1.9 |
| Aminomethyltransferase/Glycine cleavage system H protein | GcsH:GcsT | 2.9 | 5 | 0.2 | 5.4 | 5.4 |
| Histidine kinase/response regulator | KdpD:CheY (T. maritima) | 95.4 | 78 | 0.72 | 2.1 | 2.0 |
| Ubiquinol oxidase | CyoB:CyoA | 1.0 | 11 | 0.55 | 1.8 | 1.2 |
| Outer membrane usher protein/ Chaperone protein | FimD:FimC | 3.6 | 6 | 0.83 | 3.2 | 3.0 |
| Molybdopterin synthase | MoaD:MoaE | 3.6 | 8 | 1.0 | 4.4 | 4.1 |
| Methionine transporter complex | MetN:MetI | 1.9 | 14 | 0.86 | 1.5 | 1.2 |
| Dihydroxyacetone kinase | DhaL:DhaK | 1.4 | 12 | 0.42 | 6.7 | 2.4 |
| Vitamin B12 uptake system | BtuC:BtuF | 3.2 | 5 | 0.6 | 2.8 | 2.8 |
| Vitamin B12 uptake system | BtuC:BtuD | 9.8 | 21 | 0.88 | 1.1 | 0.9 |
| ATP synthase γ and ε subunits | AtpE:AtpG | 2.9 | 15 | 0.53 | 1.4 | 1.4 |
| IIA-IIB complex of the N,N'-diacetylchitobiose (Chb) transporter | PtqA:PtqB | 3.1 | 5 | 0.2 | 7.2 | 5.5 |
| 30 S Ribosomal proteins | RS3:RS14 | 1.4 | 11 | 0.91 | 1.1 | 1.1 |
| Succinatequinone oxido-reductase flavoprotein/ iron-sulfur subunits | SdhB:SdhA | 3.0 | 8 | 0.62 | 1.4 | 1.4 |
| 30 S Ribosomal proteins | RS10:RS14 | 1.2 | 6 | 1.0 | 5.3 | 2.5 |

†Number of non-redundant sequences in concatenated alignment normalized by alignment length.
‡Inter-ECs with EVcomplex score ≥0.8.
§True Positive rate for inter-ECs above score threshold.
#iRMSD positional deviation of model from known structure, for docked model with best HADDOCK score.
¶Lowest iRMSD observed across all models.

all 100 computed 3D models (**Table 1**, **Supplementary file 3** and 'Supplementary data'). The top 3 inter-EC residue pairs (K136-E108, A128-L105, and E74-R124, MetI-MetN respectively) constitute a residue network coupling the ATP-binding pocket of MetN to the membrane transporter MetI. This network calculated from the sequence alignment corresponds to residues identified experimentally that couple ATP hydrolysis to the open and closed conformations of the MetI dimer (**Johnson et al., 2012**) (**Figure 4A**). The vitamin B12 transporter (BtuC) belongs to a different structural class of ABC transporters, but also uses ATP hydrolysis via an interacting ATPase (BtuD). The top 5 inter-ECs co-locate the L-loop of BtuC close to the Q-loop ATP-binding domain of the ATPase, hence coupling the transporter with the ATP hydrolysis state in an analogous way to MetI-MetN. The identification of these coupled residues across the different subunits suggests that EVcomplex identifies not only residues close in space, but also particular pairs that are constrained by the transporter function of these complexes (**Kadaba et al., 2008**; **Johnson et al., 2012**).

The ATP synthase ε and γ subunit complex provides a challenge to our approach, since the ε subunit can take different positions relative to the γ subunit, executing the auto-inhibition of the enzyme by dramatic conformational changes (**Cingolani and Duncan, 2011**). In a real-world scenario, where we might not know this a priori, there may be conflicting constraints in the evolutionary record corresponding to the different positions of the flexible portion of ε subunit. EVcomplex accurately predicts 6 of the top 10 inter-EC pairs (within 8 Å in the crystal structure 1fs0 (**Rodgers and Wilce, 2000**) or 3oaa (**Cingolani and Duncan, 2011**)), with the top 2 inter-ECs εA45-γL215 and εA40-γL207 providing contact between the subunits along an inter-protein beta sheet. The location of the C-terminal helices of the ε subunit is significantly different across 3 crystal structures (PDB IDs: 1fs0 [**Rodgers and Wilce, 2000**],

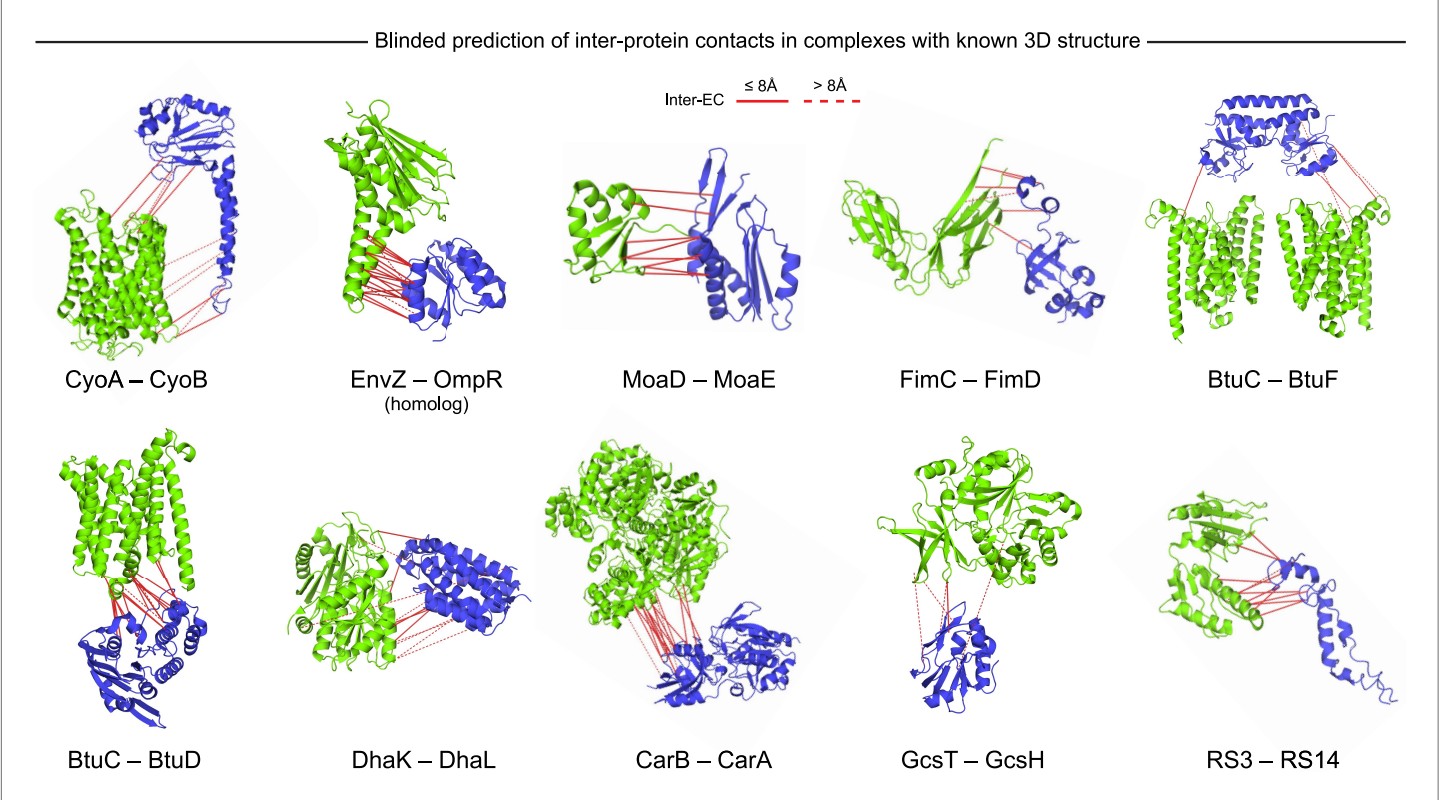

**Figure 3**. Blinded prediction of evolutionary couplings between complex subunits with known 3D structure. Inter-ECs with EVcomplex score ≥0.8 on a selection of benchmark complexes (monomer subunits in green and blue, inter-ECs in red, pairs closer than 8 Å by solid red lines, dashed otherwise). The predicted inter-ECs for these ten complexes were then used to create full 3D models of the complex using protein–protein docking. For the fifteen complexes for which 3D structures were predicted using docking, energy funnels are shown in *Figure 3—figure supplement 1*.

The following figure supplement is available for figure 3:

**Figure supplement 1**. Comparison of Interface RMSD to HADDOCK score.

1aqt [*Uhlin et al., 1997*], 3oaa [*Cingolani and Duncan, 2011*]). The top ranked intra-ECs support the conformation seen in 1aqt, with the C-terminal helices packed in an antiparallel manner and tucked against the N-terminal beta barrel (*Figure 4B*, green circles) and do not contain a high ranked evolutionary trace for the extended helical contact to the γ subunit seen in 1fs0 or 3oaa (*Figure 4B*, gray box). Docking with the top inter-ECs results in models with 1.4 Å backbone iRMSDs to the crystal structure for the interface between the N-terminal domain of the ε subunit and the γ subunit (*Table 1*, *Supplementary file 3*). εD82 and γR222 connect the ε-subunit via a network of 3 high-scoring intra-ECs between the N- and C-terminal helices to the core of the F1 ATP synthase. In summary, these examples suggest that inter-protein evolutionary couplings can provide residue relationships across the proteins that could aid identification of functional coupling pathways, in addition to obtaining 3D models of the complex.

## De novo prediction of unknown complexes

### Prediction of interactions for 32 protein pairs with high-scoring evolutionary couplings

A total of 82 protein complexes with unknown 3D structure of the interaction that satisfy the conditions for the current approach, i.e., have sufficient sequences and are close in all genomes, were predicted using EVcomplex (all residue–residue *inter*-protein evolutionary couplings scores are available in 'Supplementary data'). 32 of these have high EVcomplex scores with at least one predicted contact (*Figure 5*, *Figure 5—figure supplement 1 and 2*, and *Supplementary file 4*). Analysis of the inter-EC predictions for known 3D complex structures shows that protein pairs with more high-scoring ECs

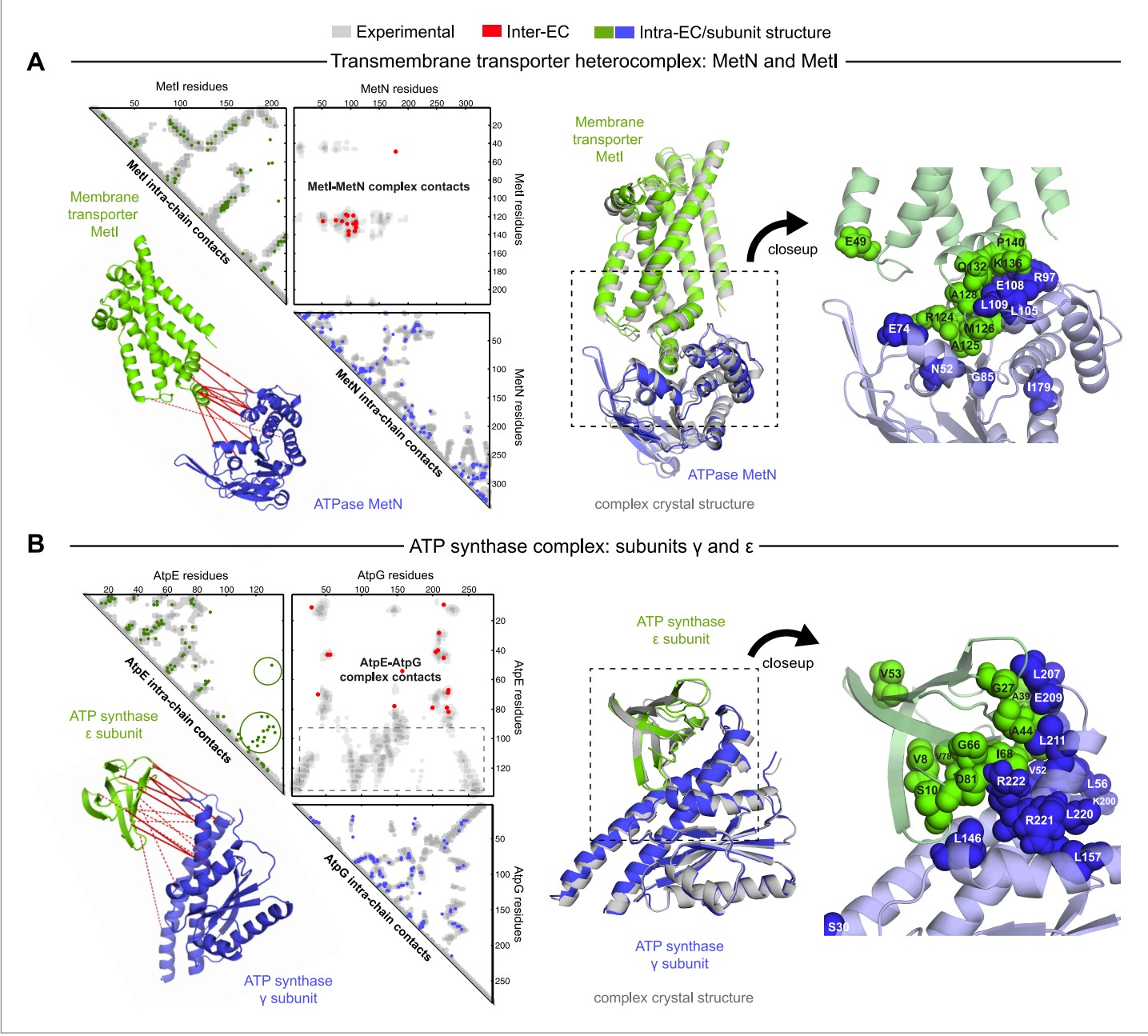

**Figure 4**. Evolutionary couplings give accurate 3D structures of complexes. EVcomplex predictions and comparison to crystal structure for (**A**) the methionine-importing transmembrane transporter heterocomplex MetNI from *E. coli* (PDB: 3tui) and (**B**) the gamma/epsilon subunit interaction of *E. coli* ATP synthase (PDB: 1fs0). Left panels: complex contact map comparing predicted inter-ECs with EVcomplex score ≥0.8 (red dots, upper right quadrant) and intra-ECs (up to the last chosen inter-EC rank; green and blue dots, top left and lower right triangles) to close pairs in the complex crystal (dark/mid/light gray points for minimum atom distance cutoffs of 5/8/12 Å for inter-subunit contacts and dark/mid gray for 5/8 Å within the subunits). Inter-ECs with an EVcomplex score ≥0.8 are also displayed on the spatially separated subunits of the complex (red lines on green and blue cartoons, couplings closer than 8 Å in solid red lines, dashed otherwise, lower left). Right panels: superimposition of the top ranked model from 3D docking (green/blue cartoon, left) onto the complex crystal structure (gray cartoon) and close-up of the interface region with highly coupled residues (green/blue spheres).

(EVcomplex score ≥0.8) have a higher proportion of true positives (*Figure 2D*). Hence, the protein complexes in the set of unknown structures with more high-scoring inter-ECs are most likely to have predicted ECs that indicate residue pairs close in 3D (column Q, *Supplementary file 2*, the exact pairs can be found in *Supplementary file 4*). Three examples of predictions with multiple high-scoring inter-ECs include MetQ-MetI, UmuD-UmuC, and DinJ–YafQ. The top 15 inter-ECs between MetQ and MetI are

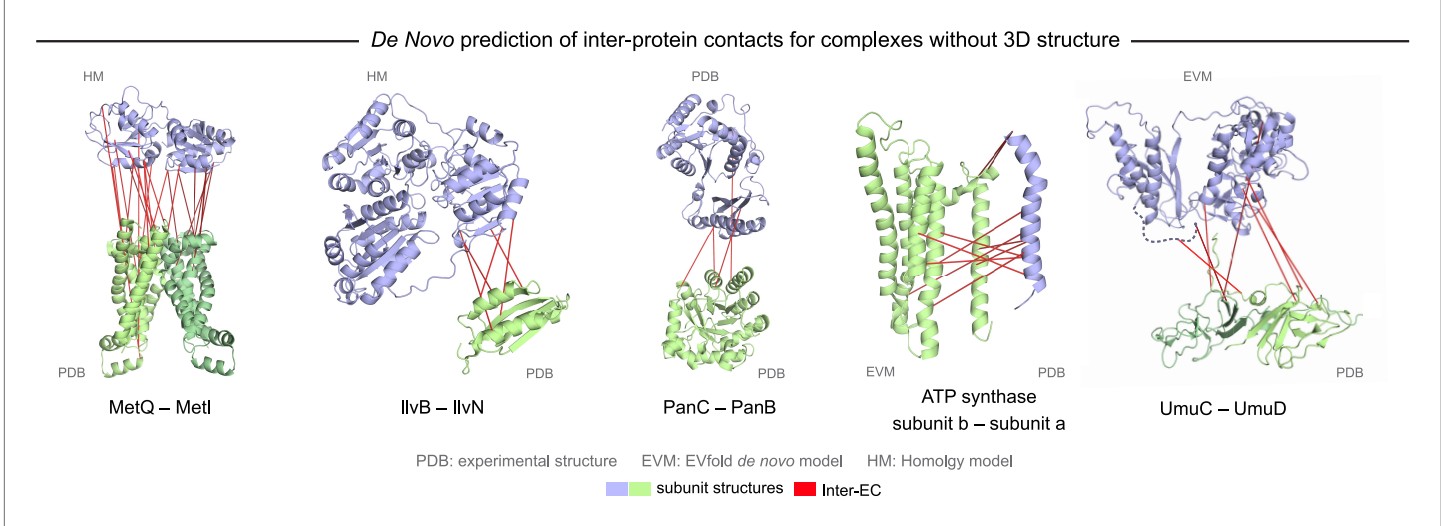

**Figure 5**. Evolutionary couplings in complexes of unknown 3D structure. Inter-ECs for five de novo prediction candidates without *E. coli* or interaction homolog complex 3D structure (Subunits: blue/green cartoons; inter-ECs with EVcomplex score ≥0.8: red lines). For complex subunits which homomulti-merize (light/dark green cartoon), inter-ECs are placed arbitrarily on either of the monomers to enable the identification of multiple interaction sites. Contact maps for all complexes with unsolved structures are provided in **Figure 5—figure supplement 1 and 2**. Left to right: (1) the membrane subunit of methionine-importing transporter heterocomplex MetI (PDB: 3tui) together with its periplasmic binding protein MetQ (Swissmodel: P28635); (2) the large and small subunits of acetolactate synthase IlvB (Swissmodel: P08142) and IlvN (PDB: 2lvw); (3) panthotenate synthase PanC (PDB: 1iho) together with ketopantoate hydroxymethyltransferase PanB (PDB: 1m3v); (4) subunits a and b of ATP synthase (model for a subunit a predict with EVfold-membrane, PDB: 1b9u for b subunit), for detailed information see **Figure 6**; and (5) the complex of UmuC (model created with EVfold) with one possible conforma-tion of UmuD (PDB: 1i4v) involved in DNA repair and SOS mutagenesis. For alternative UmuD conformation, see **Figure 5—figure supplement 3**.

The following figure supplements are available for figure 5:

**Figure supplement 1**. Contact maps of all complexes without solved 3D structure with at least one inter-ECs above EVcomplex score of 0.8.

**Figure supplement 2**. Contact maps of all complexes without solved 3D structure with at least one inter-ECs above EVcomplex score of 0.8 (2).

**Figure supplement 3**. Details of the predicted UmuCD interaction residues.

from one interface of MetQ to the MetI periplasmic loops, or the periplasmic end of the helices, consistent with the known binding of MetQ to MetI in the periplasm.

The UmuD and UmuC complex is induced in the stress/SOS response facilitating the cleavage of UmuD to UmuD' (between C24 and G25) to form UmuD'$_2$ which then interacts with UmuC (DNA pol-ymerase V) in order to copy damaged DNA (**Beuning et al., 2006**). The truncated dimer form (UmuD'$_2$) has at least two contrasting conformations where the N-terminal arm is placed on opposite sides of the dimer in one conformation or in close proximity in the alternative (**Figure 5—figure supplement 3**). For 6/7 ECs above the score threshold, residues in UmuD predicted to interface with UmuC are co-located on one face of the dimer. Two residues (Y33, I38) are located in the N-terminal arm of UmuD that, after cleavage of the 24 N-terminal amino acids, may become available for binding UmuC. Since UmuD switches functions after this cleavage and can then bind UmuC, these inter-ECs may iden-tify the critical residues for translesion synthesis function (**Beuning et al., 2006**). Although the ECs from this UmuD arm to UmuC involve residues in two separate domains of UmuC (S 415 and Y 74), intra-monomer evolutionary couplings predict that these residues are close in UmuC (**Figure 5—figure supplement 3A**, black rectangles). The relative positions of the contacting residues within each monomer therefore support the plausibility of the accuracy of the interaction interface.

Whilst this manuscript was in review, the 3D structure of the previously unsolved biofilm toxin/antitoxin DinJ–YafQ complex was published (PDB: 3mlo (**Liang et al., 2014**)), showing the intertwining of subu-nits in a heterotetrameric complex. 17/19 predicted EC residue pairs are within 8 Å in this 3D structure (**Supplementary file 4** and 'Supplementary data'). In general, the agreement between our de novo predicted inter-protein ECs and available experimental data serves as a measure of confidence for the

predicted residue pair interactions and suggests that EVcomplex can be used to reveal 3D structural details of yet unsolved protein complexes given sufficient evolutionary information.

## EVcomplex predicts interacting protein pairs in a large complex

To investigate whether the EVcomplex score can also distinguish between interacting and non-interacting pairs of proteins, we use the *E. coli* ATP synthase complex as a test case. The ATP synthase structure is of wide biological interest (reviewed in *Walker, 2013*) with a remarkable 3D structural arrangement, but completion of all aspects of the 3D structure has remained experimentally challenging (*Baker et al., 2012*) (*Figure 6A*). As a demonstration exercise, we calculated evolutionary couplings for all 28 possible pair combinations of different ATP synthase subunits (centered around the *E. coli* ATP synthase) and transformed the ECs into EVcomplex scores for all inter-protein residue pairs (experimentally determined stoichiometry: α3β3γδεab2c10, *Supplementary file 5* and 'Supplementary data'). Using the default EVcomplex score threshold of 0.8 to discriminate between interacting and non-interacting pairs of subunits, 24 of the 28 possible interactions between the subunits are correctly classified as interacting or non-interacting. The four incorrect predictions (namely: ε and c, γ and c, ε and β, b and β, for which there is some experimental evidence) are not identified as interacting using the 0.8 EVcomplex threshold. Choosing a threshold lower than 0.8 does identify 2 of these as interacting but also introduces new false positives. The ε and β interaction in the crystal structure 3oaa (*Cingolani and Duncan, 2011*) is a special case in that it involves a highly extended conformation of the last two helices of the ε subunit that reach up into the enzyme making contacts with the β subunit. The false negative EVcomplex score for this pair could be a result of the transience of their interaction or reflect a more general problem of lack of conservation of this interaction across the aligned proteins from different species. In total 80% of the interacting residue pairs in the known 3D structure parts of the synthase complex (7 pairs of subunits) are correctly predicted (threshold: 10 Å minimum atom distance between two residues). This exercise of predicting the presence or absence of interaction between any two proteins indicates the potential of the EVcomplex method in helping to elucidate protein–protein interaction networks from evolutionary sequence co-variation and identify interacting subunits of large macromolecular complexes.

## EVcomplex predicts details of subunit interactions in ATP synthase

While much of the 3D structure of ATP synthase is known (*Walker, 2013*), the details of interactions between the a-, b-, and c-subunits have not yet been determined by crystallography. We analyse the details in these interactions, as the EVcomplex scores between these subunits are substantial (*Figure 6B*). We are fortunately able to provide a missing piece for this analysis, the unknown structure of the membrane-integral penta-helical a-subunit, using our previously described method for de novo 3D structure prediction of alpha-helical transmembrane proteins (*Hopf et al., 2012*). To our knowledge, there are no experimentally determined atomic resolution structures of the a-subunit of ATP synthase. A 3D model of the a-subunit is from 1999 (1c17(*Rastogi and Girvin, 1999*)) and was computed using five helical–helical interactions that were inferred from second suppressor mutation experiments, and then imposed as distance restraints for TMH2-5, revealing a four helical bundle (with no information for TMH1). Later, cross-linking experiments (*Schwem and Fillingame, 2006*) identified contacting residues from all pairs of helical combinations of TM2-TM5 (6 pairs), supporting the earlier 4 helical bundle topology. 7 of the 8 cross-linked pairs are either exactly the same pair (L120-I246) or adjacent to many pairs in the top L intra a-subunit evolutionary couplings (ECs).

In fact, the helix packing arrangement in the predicted structure of the a-subunit is consistent with the topology suggested on the basis of crosslinking studies (*Long et al., 2002*; *DeLeon-Rangel et al., 2003*; *Fillingame and Steed, 2014*), including the lack of contacts for transmembrane helix 1 with the other 4 helices ('Supplementary data').

The top inter-protein EC pair between subunits a and b, aK74–bE34, coincides with experimental crosslinking evidence of the interaction of aK74 with the b-subunit and the position of E34 of the b subunit emerging from the membrane on the cytoplasmic side (*Long et al., 2002*; *DeLeon-Rangel et al., 2003*). Indeed, 6 of the 13 high score ECs are in the same region as the experimental crosslinks, for instance between the cytoplasmic loop between the first two helices of the a-subunit and the b-subunit helix as it emerges from the membrane bilayer (*DeLeon-Rangel et al., 2013*), or between a239V in TM helix 5 and bL16 (*Figure 6C*, *Figure 6—figure supplement 1*, *Supplementary file 6*). Additionally, the top EC between the a- and c-subunits (aG213 – cM65) lies close to the functionally

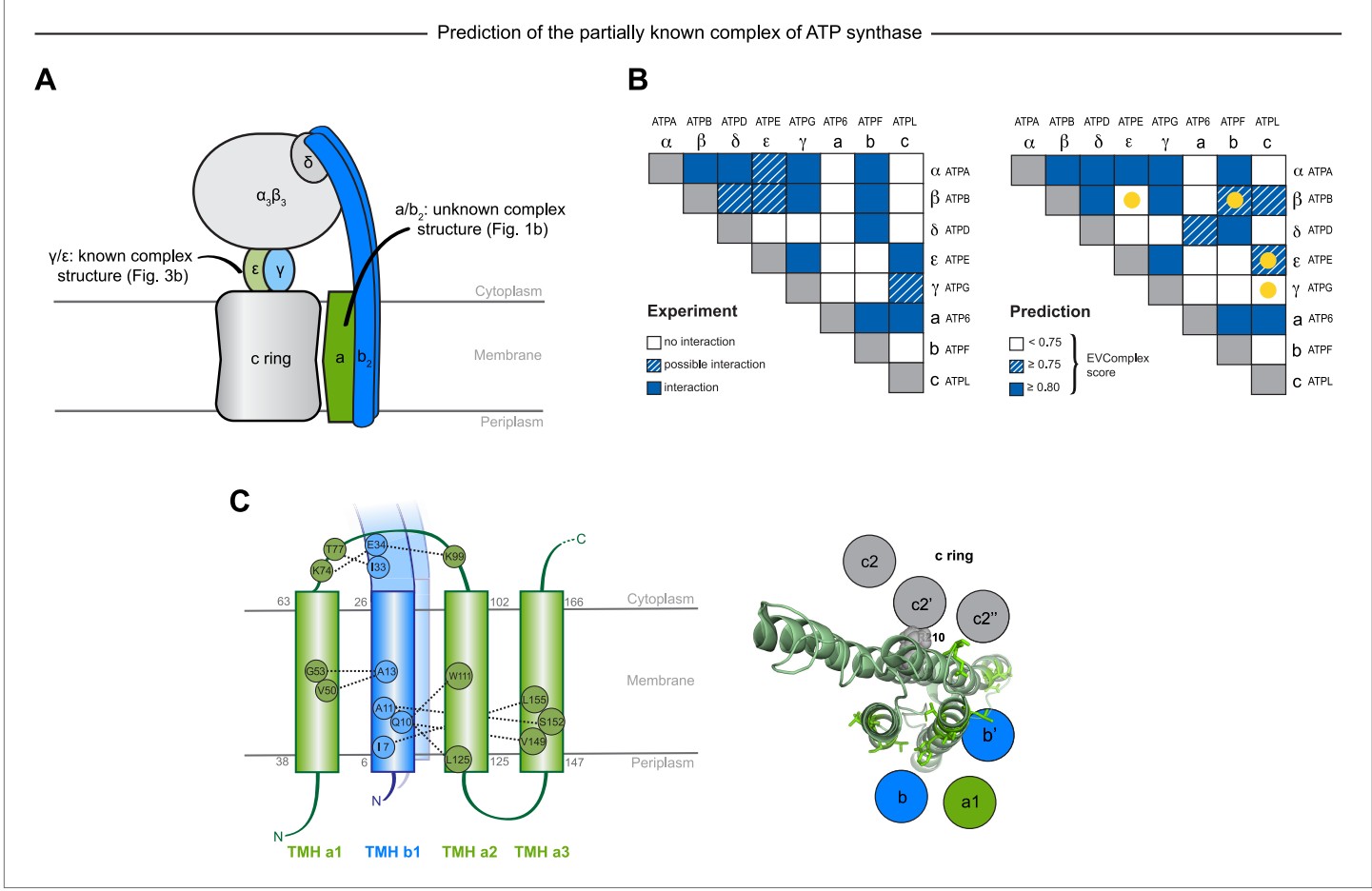

**Figure 6**. Predicted interactions between the a-, b-, and c-subunits of ATP synthase. (**A**) The a- and b- subunits of *E. coli* ATP synthase are known to interact, but the monomer structure of subunits a and b and the structure of their interaction in the complex are unknown. (**B**) EVcomplex prediction (right matrix) for ATP synthase subunit interactions compared to experimental evidence (left matrix), which is either strong (left, solid blue squares) or indicative (left, crosshatched squares). Interactions that have experimental evidence, but are not predicted at the 0.8 threshold are indicated as yellow dots. (**C**) Left panel: residue detail of predicted residue–residue interactions (dotted lines) between subunit a and b (residue numbers at the boundaries of transmembrane helices in gray). Right panel: proposed helix–helix interactions between ATP synthase subunits a (green), b (blue, homodimer), and the c ring (gray). The proposed structural arrangement is based on analysis of the full map of inter-subunit ECs with EVcomplex score ≥0.8 (***Figure 6— figure supplement 1***).

The following figure supplement is available for figure 6:

**Figure supplement 1**. Contact map of predicted ECs in the ATPsynthase a and b subunits.

critical aR210–cD61 interaction (***Dmitriev et al., 1999***) on the same helical faces of the respective subunits (***Figure 6C***). This prediction of missing aspects of subunit interactions may help in the design of targeted experiments to complete the understanding of the intricate molecular mechanism of the ATP synthase complex.

## Discussion

A primary limitation of our current approach is its dependence on the availability of a large number of evolutionarily related sequences. If a protein interaction is conserved across enough sequenced genomes, using a single pair per genome can give accurate predictions of the interacting residues. However, if the protein pair is present in limited taxonomic branches, there may be insufficient sequences at any given time to make confident predictions. A solution to this could be to include multiple paralogs of the interacting proteins from each genome, but this requires correct pairing of the interaction partners, which is in general hard to ascertain. In addition, details of interactions may have

diverged for paralogous pairs. Hence, in this current version of the method, we have imposed a genome distance requirement across all genomes for all homolog pairs in order to be less sensitive to these complications.

As the need to use genome proximity to pair sequences becomes less important with the increasing availability of genome sequences, there will be a dramatic increase in the number of interactions that can be inferred from evolutionary couplings, including those unique to eukaryotes. With currently available sequences (May 2014 release of the UniProt database), EVcomplex is able to provide information for about 1/10th of the known 3000 protein interactions in the *E. coli* genome. Once there are ~10,000 bacterial genome sequences of sufficient diversity, one would have enough information to test each potentially interacting pair of homologs for evidence of interaction and, given sufficiently strong evolutionary couplings, infer the 3D structure of each protein–protein pair, as well as of complexes with more than two proteins. For any set of species, e.g., vertebrates or mammals, one can imagine guiding sequencing efforts to optimize species diversity to facilitate the extraction of evolutionary couplings. This can open the doors for more comprehensive and more rapid determination of approximate 3D structures of proteins and protein complexes, as well as for the elucidation in molecular detail of the most strongly evolutionarily constrained interactions, pointing to functional interactions.

Determining the three-dimensional models of complexes from the predicted contacts was successful in many of the tested instances. Using minimal computing resources and a small number of inter-EC-derived contacts, low interface positional RMSDs relative to experimental structures can be achieved. However, a significant number of proteins exist as homomultimers within larger complexes. To determine models of these complexes one must deconvolute homomultimeric inter-ECs from the intra-protein signal, which is an important technical challenge for future work.

The analysis of subunit interactions in ATP synthase in this work is a 'proof of principle' study showing that methods such as EVcomplex can determine which proteins interact with each other at the same time as specific residue pair couplings across the proteins (as also shown in the work by the Baker lab on ribosomal protein interactions (*Ovchinnikov et al., 2014*)). Understanding the networks of protein interactions is of critical interest in eukaryotic systems, such as networks of protein kinases, GPCRs, or PDZ domain proteins. An understanding of the distributions of interaction specificities is of high interest to many fields. Although we do not know how well our evolutionary coupling approach will handle less obligate interactions, results on the two-component signaling system (histidine kinase/ response regulator) both here and in other work (*Skerker et al., 2008*; *Weigt et al., 2009*) suggest optimism.

The approximately scale-free EVcomplex score is a heuristic based on the distribution of raw EC scores from the statistical model, their dependence on sequence alignment depth and the length of the concatenated sequences. The score provides a simple way of accounting for these dependencies such that a uniform threshold, say 0.8, can be used for any protein pair with the expectation of reasonably accurate predictions. Since cutoff thresholds can be useful but overly sharp, we recommend investigating predicted contacts below the threshold used in this work, especially where there is independent biological knowledge to validate the predictions.

The work presented here is in anticipation of a genome-wide exploration and, as a proof of principle, shows the accurate prediction of inter-protein contacts in many cases and their utility for the computation of 3D structures across diverse complex interfaces. As with single protein (intra-EC) predictions, evolutionarily conserved conformational flexibility and oligomerization can result in more than one set of contacts that must be de-convoluted. Can evolutionary information help to predict the details and extent for each complex? A key challenge will be the development of algorithms that can disentangle evolutionary signals caused by alternative conformations of single complexes, alternative conformations of homologous complexes, and effectively deal with false positive signals. Taken together, these issues highlight fruitful areas for future development of evolutionary coupling methods.

Despite conditions for the successful de novo calculation of co-evolved residues, the method described here may accelerate the exploration of the protein–protein interaction world and the determination of protein complexes on a genome-wide scale at residue level resolution. The use of co-evolutionary analysis in computational models to determine protein specificity and promiscuity, co-evolutionary dynamics and functional drift will open up exciting future research questions.

# Materials and methods

## Selection of interacting protein pairs for co-evolution calculation

The candidate set of complexes for testing and de novo prediction was derived starting from a data set of binary protein–protein interactions in *E. coli* including yeast two-hybrid experiments, literature-curated interactions and 3D complex structures in the PDB (*Rajagopala et al., 2014*). Three complexes not contained in the list were added based on our analysis of other subunits in the same complex, namely BtuC/BtuF, MetI/MetQ, and the interaction between ATP synthase subunits a and b. Since our algorithm for concatenating multiple sequence pairs per species assumes the proximity of the interacting proteins on the respective genomes of each species (See below), we excluded any complex with a gene distance >20 from further analysis. The gene distance is calculated as the number of genes between the interacting partners based on an ordered list of genes in the *E. coli* genome obtained from the UniProt database. The resulting list of pairs (~350) was then filtered for pseudo-homomultimeric complexes based on the identification of Pfam domains in the interacting proteins (330). All remaining complexes with a known 3D structure (as summarized in *Rajagopala et al., 2014*) or a homologous interacting 3D structure (93) (identified by intersecting the results of HMMER searches against the PDB for both monomers) were used for evaluating the method, while complexes without known structure (236) were assigned to the de novo prediction set (*Figure 1—figure supplement 1*). The set with protein complexes of known 3D structure was further filtered for structures that only cover fragments (<30 amino acids) of one or both of the monomers and structures with very low resolution (>5 Å), which led to the re-assignment of Ribonucleoside-diphosphate reductase 1 (complex_002), Type I restriction-modification enzyme EcoKI (complex_012), RpoC/RpoB (complex_041), RL11/Rl7 (complex_165), the ribosome with SecY (complex_226, complex_250, and complex_255), and RS3/RS (complex_254) to the set of unknown complexes. Large proteins were run with the specific interacting domains informed by the known 3D structure, when the full sequence was too large for the number of retrieved sequences (for domain annotation see 'Supplementary data').

This set could serve as a benchmark set for future development efforts in the community.

## Multiple sequence alignments

Each protein from all pairs in our data set was used to generate a multiple sequence alignment (MSA) using jackhmmer (*Johnson et al., 2010*) to search the UniProt database (*UniProt Consortium, 2014*) with 5 iterations. To obtain alignments of consistent evolutionary depths across all the proteins, a bit score threshold of 0.5 * monomer sequence length was chosen as homolog inclusion criterion (-incdomT parameter), rather than a fixed *E*-value threshold which selects for different degrees of evolutionary divergence based on the length of the input sequence.

In order to calculate co-evolved residues across different proteins, the interacting pairs of sequences in each species need to be matched. Here, we assume that proteins in close proximity on the genome, e.g., on the same operon, are more likely to interact, as in the methods used previously matching histidine kinase and response regulator interacting pairs (*Skerker et al., 2008*; *Weigt et al., 2009*) ('Supplementary data'). We retrieved the genomic locations of proteins in the alignments and concatenated pairs following 2 rules: (i) the CDS of each concatenated protein pair must be located on the same genomic contig (using ENA (*Pakseresht et al., 2014*) for mapping) and (ii) each pair must be the closest to one another on the genome, when compared to all other possible pairings in the same species. The concatenated sequence pairs were filtered based on the distribution of genomic distances to exclude outlier pairs with high genomic distances of more than 10k nucleotides ('Supplementary data'). Alignment members were clustered together and reweighted if 80% or more of their residues were identical (thus implicitly removing duplicate sequences from the alignment). *Supplementary file 1 and 2* report the total number of concatenated sequences, the lengths, and the effective number of sequences remaining after down-weighting in the evaluation and de novo prediction set, respectively.

## Computation of evolutionary couplings

Inter- and intra-ECs were calculated on the alignment of concatenated sequences using a global probability model of sequence co-evolution, adapted from the method for single proteins (*Marks et al., 2011*; *Morcos et al., 2011*; *Hopf et al., 2012*) using a pseudo-likelihood maximization (PLM) (*Balakrishnan et al., 2011*; *Ekeberg et al., 2013*) rather than mean field approximation to calculate the coupling parameters. Columns in the alignment that contain more than 80% gaps were excluded

and the weight of each sequence was adjusted to represent its cluster size in the alignment thus reducing the influence of identical or near-identical sequences in the calculation. For the evaluation set, we can then compare the predicted ECs for both within and between the proteins/domains to the crystal structures of the complexes (for contact maps and all EC scores, see 'Supplementary data').

## Definition of a scale-free score for the assessment of interactions

In order to estimate the accuracy of the EC prediction, we evaluate the calculated inter-ECs based on the following observations: (1) most pairs of positions in an alignment are not coupled, i.e., have an EC score close to zero, and tend to be distant in the 3D structure; (2) the background distribution of EC scores between non-coupled positions is approximately symmetric around a zero mean; and (3) higher-scoring positive score outliers capture 3D proximity more accurately than lower-scoring outliers (See also *Figure 2*). The width of the (symmetric) background EC score distribution can be approximated using the absolute value of the minimal inter-EC score. The more a positive EC score exceeds the noise level of background coupling, the more likely it is to reflect true co-evolution between the coupled sites. For each inter-protein pair of sites i and j with pair coupling strength $EC_{inter}(i, j)$, we therefore calculate a raw reliability score ('pair coupling score ratio', *Figure 2B*) defined by

$$Q_{inter}^{raw}(i, j) = \frac{EC_{inter}(i, j)}{\left| \min_{i,j}\left( EC_{inter}(i, j) \right) \right|} \tag{1}$$

Since the accuracy of evolutionary couplings critically depends both on the number and diversity of sequences in the input alignment and the size of the statistical inference problem (*Marks et al., 2011*; *Morcos et al., 2011*; *Jones et al., 2012*), we incorporate a normalization factor to make the raw reliability score comparable across different protein pairs. The normalized EVcomplex score is defined as

$$EVcomplex\text{-}Score(i, j) = \frac{Q_{inter}^{raw}(i, j)}{1 + \left( \frac{N_{eff}}{L} \right)^{-\frac{1}{2}}} \tag{2}$$

where $N_{eff}$ is the effective number of sequences in the alignment after redundancy reduction, and L (total number of residues) is the length of the concatenated alignment. Previous work on single proteins has shown that the method requires a sufficient number of sequences in the alignment to be statistically meaningful. We thus filter for sequence sufficiency requiring $N_{eff}/L > 0.3$ (*Table 1*, *Supplementary files 1 and 2*). Predictions of coupled residues in the evaluation set were evaluated against their residue distances in known structures of protein pairs (*Rajagopala et al., 2014*) (See *Supplementary file 7*) in order to determine the precision of the method.

To interpret the EVcomplex prediction of interaction between subunits a and b of the ATP synthase as well as UmuC and UmuD, individual monomer models were built de novo for the structurally unsolved a-subunit of ATP synthase and UmuC using the EVfold pipeline as previously published (*Marks et al., 2011*; *Hopf et al., 2012*). In both cases coupling parameters were calculated using PLM (*Balakrishnan et al., 2011*; *Ekeberg et al., 2013*) and sequences were clustered and weighted at 90% sequence identity (the resulting models are provided in 'Supplementary data').

## Prediction of interactions in a set of subunits

Following this same protocol EVcomplex scores were calculated for all possible 28 combinations of the 8 *E. coli* ATP synthase $F_0$ and $F_1$ subunits. Since we want to compare the computational predictions to some 'ground truth', as with the complexes for the rest of the manuscript, we used known 3D structures of the ATP synthase complex to assign whether or not the subunits interact (PDB: 3oaa, 1fs0, 2a7u; *Supplementary file 7*). Since we are also determining whether the subunits interact, not necessarily knowing full atomic detail residue interactions, we included subunit interactions that have been inferred from cryo-EM, crosslinking or other experiments, but do not necessarily have a crystal structure. These are represented as solid blue boxes, if the interaction is well established (*DeLeon-Rangel et al., 2013*; *Schulenberg et al., 1999*; *Brandt et al., 2013*; *McLachlin and Dunn, 2000*), or crosshatched blue if there is a lack of consensus in the community (*Figure 6B*, left panel).

For each possible interaction the EVcomplex score of the highest ranked inter-EC was considered as a proxy for the likelihood of interaction. Pairs with scores above 0.8 are considered likely to interact,

between 0.75 and 0.8 weakly predicted, while interactions with scores below 0.75 are rejected as possible complexes (blue boxes, blue crosshatched, and white respectively in right panel of *Figure 6B*, and 'Supplementary data').

## Computation of 3D structure of complexes

A diverse set of 15 complexes was chosen from the 22 in the evaluation set that had at least 5 couplings above a complex score of 0.8 and was subsequently docked (*Supplementary file 3*). Proteins that have been crystallized together in a complex could bias the results of the docking, as they have complementary positions of the surface side chains. Therefore, where possible we used complexes that had a solved 3D structure of the unbound monomer, namely GcsH/GcsT, CyoA, FimC, DhaL, AtpE, PtqA/PtqB, RS10, and HK/RR, and in all other cases the side chains of the monomers were randomized either by using SCWRL4 (*Krivov et al., 2009*) or restrained minimization with Schrodinger Protein Preparation Wizard (*Sastry et al., 2013*) before docking. For ubiquinol oxidase (complex_054) the unbound structure of subunit 2 (CyoA) only covers the COX2 domain. In this case docking was performed using this unbound structure plus an additional run using the bound complex structure with perturbed side chains.

We used HADDOCK (*Dominguez et al., 2003*), a widely used docking program based on ARIA (*Linge et al., 2003*) and the CNS software (*Brunger, 2007*) (Crystallography and NMR System), to dock the monomers for each protein pair with all inter-ECs with an EVcomplex score of 0.8 or above implemented as distance restraints on the α-carbon atoms of the backbone.

Each docking calculation starts with a rigid-body energy minimization, followed by semi-flexible refinement in torsion angle space, and ends with further refinement of the models in explicit solvent (water). 500/100/100 models generated for each of the 3 steps, respectively. All other parameters were left as the default values in the HADDOCK protocol. Each protein complex was run using predicted ECs as unambiguous distance restraints on the Cα atoms ($d_{eff}$ 5 Å, upper bound 2 Å, lower bound 2 Å; input files available in 'Supplementary data'). As a negative control, each protein complex was also docked using center of mass restraints alone (ab initio docking mode of HADDOCK) (*de Vries et al., 2007*) and in this case generating 10,000/500/500 models.

Each of the generated models is scored using a weighted sum of electrostatic ($E_{elec}$) and van der Waals ($E_{vdw}$) energies complemented by an empirical desolvation energy term ($E_{desolv}$) (*Fernandez-Recio et al., 2004*). The distance restraint energy term was explicitly removed from the equation in the last iteration (Edist3 = 0.0) to enable comparison of the scores between the runs that used a different number of ECs as distance restraints.

## Comparison of predicted to experimental structures

All computed models in the docked set were compared to the cognate crystal structures by the RMSD of all backbone atoms at the interface of the complex using ProFit v.3.1 (http://www.bioinf.org.uk/software/profit/). The interface is defined as the set of all residues that contain any atom <6 Å away from any atom of the complex partner. For the AtpE–AtpG complex, we excluded the 2 C-terminal helices of AtpE as these helices are mobile and take many different positions relative to other ATP synthase subunits (*Cingolani and Duncan, 2011*). Similarly, since the DHp domain of histidine kinases can take different positions relative to the CA domain, the HK-RR complex was compared over the interface between the DHp domain alone and the response regulator partner. In the case of the unbound ubiquinol oxidase docking results, only the interface between COX2 in subunit 2 and subunit 1 was considered. Accuracy of the computed models with EC restraints was compared with computed models with center of mass restraints alone (negative controls) (*Figure 3—figure supplement 1*, *Supplementary file 3*).

Data analysis was conducted primarily using IPython notebooks (*Perez and Granger, 2007*). A webserver and all data are available at EVcomplex.org and the Dryad Digital Repository (*Hopf et al., 2014*).

## Supplementary Data

(Available at http://doi.org/10.5061/dryad.6t7b8 [*Hopf et al., 2014*])

 Supplementary data 1: Concatenated alignments for complexes predicted in this work
 Supplementary data 2: Genome distance distribution of concatenated sequences per alignment
 Supplementary data 3: EVcomplex predictions for evaluation and de novo set
 Supplementary data 4: Docking input files and top 10 predicted models for evaluation set
 Supplementary data 5: ATP synthase predictions, ATP synthase subunit a model
 Supplementary data 6: UmuC model

## Additional information

### Funding

| Funder | Grant reference number | Author |
|---|---|---|
| National Institute of General Medical Sciences | R01 GM106303 | Thomas A Hopf, Debora S Marks, Chris Sander |
| Fulbright Commission | | Charlotta P I Schärfe |

The funders had no role in study design, data collection and interpretation, or the decision to submit the work for publication.

### Author contributions

TAH, CPIS, DSM, Conception and design, Acquisition of data, Analysis and interpretation of data, Drafting or revising the article; JPGLMR, Assisted docking in Haddock; AGG, Conducted comparison of ATP synthase subunit interactions; OK, Acquisition of data, Analysis and interpretation of data; CS, Conception and design, Drafting or revising the article; AMJJB, Provided expertise on Haddock docking protocols, Drafting or revising the article

### Author ORCIDs

Thomas A Hopf, http://orcid.org/0000-0002-7476-9539
Charlotta P I Schärfe, http://orcid.org/0000-0002-6689-6423
Debora S Marks, http://orcid.org/0000-0001-9388-2281
Alexandre M J J Bonvin, http://orcid.org/0000-0001-7369-1322

## Additional files

### Supplementary files

• Supplementary file 1. Benchmark data set and results.

• Supplementary file 2. De novo prediction data set and results.

• Supplementary file 3. Docking results.

• Supplementary file 4. Predicted inter-ECs for complexes in de novo prediction data set with EVcomplex score ≥0.8.

• Supplementary file 5. ATP synthase interaction predictions.

• Supplementary file 6. Comparison of ATP synthase EVcomplex predictions of a and b subunit with cross-linking studies.

• Supplementary file 7. PDB identifiers used for comparison of predicted evolutionary couplings to known 3D structures.

### Major datasets

The following dataset was generated

| Author(s) | Year | Dataset title | Dataset ID and/or URL | Database, license, and accessibility information |
|---|---|---|---|---|
| Hopf T, Schärfe C, Rodrigues J, Green A, Sander C, Bonvin A, Marks D | 2014 | Data from: Sequence co-evolution gives 3D contacts and structures of protein complexes | http://dx.doi.org/10.5061/dryad.6t7b8 | Publicly available at Dryad Digital Repository. |

The following previously published datasets were used:

| Author(s) | Year | Dataset title | Dataset ID and/or URL | Database, license, and accessibility information |
|---|---|---|---|---|
| Thoden JB, Huang X, Raushel FM, Holden HM | 2004 | Crystal structure of E. coli carbamoyl phosphate synthetase small subunit mutant C248D complexed with uridine 5'-monophosphate | 1T36; http://dx.doi.org/10.2210/pdb1T36/pdb | Publicly available at the RCSB Protein Data Bank. |
| Borovinskaya MA, Pai RD, Zhang W, Schuwirth BS, Holton JM, Hirokawa G, Kaji H, Kaji A, Cate JHD | 2007 | Crystal structure of the bacterial ribosome from Escherichia coli in complex with gentamicin. This file contains the 50S subunit of the first 70S ribosome, with gentamicin bound. The entire crystal structure contains two 70S ribosomes and is described in remark 400 | 2QBA; http://dx.doi.org/10.2210/pdb2QBA/pdb | Publicly available at the RCSB Protein Data Bank. |
| Zhao R, Collins EJ, Bourret RB, Silversmith RE | 2001 | Crystal Structure of an E.coli Chemotaxis Protein, Chez | 1KMI; http://dx.doi.org/10.2210/pdb1KMI/pdb | Publicly available at the RCSB Protein Data Bank. |
| Borovinskaya MA, Pai RD, Zhang W, Schuwirth BS, Holton JM, Hirokawa G, Kaji H, Kaji A, Cate JHD | 2007 | Crystal structure of the bacterial ribosome from Escherichia coli in complex with gentamicin. This file contains the 30S subunit of the second 70S ribosome, with gentamicin bound. The entire crystal structure contains two 70S ribosomes and is described in remark 400 | 2QBB; http://dx.doi.org/10.2210/pdb2QBB/pdb | Publicly available at the RCSB Protein Data Bank. |
| Paz A, Vartanian AS, Fortgang EA, Abramson J, Dahlquist FW | 2012 | Crystal structure of the complex between the flagellar motor proteins FliG and FliM | 4FHR; http://dx.doi.org/10.2210/pdb4FHR/pdb | Publicly available at the RCSB Protein Data Bank. |
| Schuwirth BS, Borovinskaya MA, Hau CW, Zhang W, Vila-Sanjurjo A, Holton JM, Cate JHD | 2005 | Crystal structure of the bacterial ribosome from Escherichia coli at 3.5 A resolution. This file contains the 30S subunit of the second 70S ribosome. The entire crystal structure contains two 70S ribosomes and is described in remark 400 | 2AW7; http://dx.doi.org/10.2210/pdb2AW7/pdb | Publicly available at the RCSB Protein Data Bank. |
| Schuwirth BS, Borovinskaya MA, Hau CW, Zhang W, Vila-Sanjurjo A, Holton JM, Cate JHD | 2005 | Crystal structure of the bacterial ribosome from Escherichia coli at 3.5 A resolution. This file contains the 50S subunit of one 70S ribosome. The entire crystal structure contains two 70S ribosomes and is described in remark 400 | 2AW4; http://dx.doi.org/10.2210/pdb2AW4/pdb | Publicly available at the RCSB Protein Data Bank. |
| Borovinskaya MA, Pai RD, Zhang W, Schuwirth BS, Holton JM, Hirokawa G, Kaji H, Kaji A, Cate JHD | 2007 | Crystal structure of the bacterial ribosome from Escherichia coli in complex with ribosome recycling factor (RRF). This file contains the 30S subunit of the second 70S ribosome. The entire crystal structure contains two 70S ribosomes and is described in remark 400 | 2QBF; http://dx.doi.org/10.2210/pdb2QBF/pdb | Publicly available at the RCSB Protein Data Bank. |

| | | | |
|---|---|---|---|
| Borovinskaya MA, Pai RD, Zhang W, Schuwirth BS, Holton JM, Hirokawa G, Kaji H, Kaji A, Cate JHD | 2007 | Crystal structure of the bacterial ribosome from Escherichia coli in complex with gentamicin and ribosome recycling factor (RRF). This file contains the 50S subunit of the first 70S ribosome, with gentamicin and RRF bound. The entire crystal structure contains two 70S ribosomes and is described in remark 400 | 2QBI; http://dx.doi.org/ 10.2210/pdb2QBI/pdb | Publicly available at the RCSB Protein Data Bank. |
| Borovinskaya MA, Pai RD, Zhang W, Schuwirth BS, Holton JM, Hirokawa G, Kaji H, Kaji A, Cate JHD | 2007 | Crystal structure of the bacterial ribosome from Escherichia coli in complex with gentamicin and ribosome recycling factor (RRF). This file contains the 30S subunit of the first 70S ribosome, with gentamicin bound. The entire crystal structure contains two 70S ribosomes and is described in remark 400 | 2QBH; http://dx.doi.org/ 10.2210/pdb2QBH/pdb | Publicly available at the RCSB Protein Data Bank. |
| Borovinskaya MA, Pai RD, Zhang W, Schuwirth BS, Holton JM, Hirokawa G, Kaji H, Kaji A, Cate JHD | 2007 | Crystal structure of the bacterial ribosome from Escherichia coli in complex with gentamicin and ribosome recycling factor (RRF). This file contains the 50S subunit of the second 70S ribosome, with gentamicin and RRF bound. The entire crystal structure contains two 70S ribosomes and is described in remark 400 | 2QBK; http://dx.doi.org/ 10.2210/pdb2QBK/pdb | Publicly available at the RCSB Protein Data Bank. |
| Borovinskaya MA, Pai RD, Zhang W, Schuwirth BS, Holton JM, Hirokawa G, Kaji H, Kaji A, Cate JHD | 2007 | Crystal structure of the bacterial ribosome from Escherichia coli in complex with gentamicin and ribosome recycling factor (RRF). This file contains the 30S subunit of the second 70S ribosome, with gentamicin bound. The entire crystal structure contains two 70S ribosomes and is described in remark 400 | 2QBJ; http://dx.doi.org/ 10.2210/pdb2QBJ/pdb | Publicly available at the RCSB Protein Data Bank. |
| Zhang W, Dunkle JA, Cate JHD | 2009 | Crystal structure of the E. coli 70S ribosome in an intermediate state of ratcheting | 3I1P; http://dx.doi.org/ 10.2210/pdb3I1P/pdb | Publicly available at the RCSB Protein Data Bank. |
| Borovinskaya MA, Pai RD, Zhang W, Schuwirth BS, Holton JM, Hirokawa G, Kaji H, Kaji A, Cate JHD | 2007 | Crystal structure of the bacterial ribosome from Escherichia coli in complex with gentamicin. This file contains the 50S subunit of the second 70S ribosome, with gentamicin bound. The entire crystal structure contains two 70S ribosomes and is described in remark 400 | 2QBC; http://dx.doi.org/ 10.2210/pdb2QBC/pdb | Publicly available at the RCSB Protein Data Bank. |

| | | | | |
|---|---|---|---|---|
| Evdokimov A, Voznesensky I, Fennell K, Anderson M, Smith JF, Fisher DA | 2009 | Mercury-modified bacterial persistence regulator hipBA | 2WIU; http://dx.doi.org/10.2210/pdb2WIU/pdb | Publicly available at the RCSB Protein Data Bank. |
| Maklashina E, Iverson TM, Sher Y, Kotlyar V, Mirza O, Andrell J, Hudson JM, Armstrong FA, Cecchini G | 2005 | E. coli Quinol fumarate reductase FrdA E49Q mutation | 2B76; http://dx.doi.org/10.2210/pdb2B76/pdb | Publicly available at the RCSB Protein Data Bank. |
| Yamada K, Miyata T, Tsuchiya D, Oyama T, Fujiwara Y, Ohnishi T, Iwasaki H, Shinagawa H, Ariyoshi M, Mayanagi K, Morikawa K | 2002 | RuvA-RuvB complex | 1IXR; http://dx.doi.org/10.2210/pdb1IXR/pdb | Publicly available at the RCSB Protein Data Bank. |
| Bertero MG, Rothery RA, Weiner JH, Strynadka NCJ | 2009 | Crystal structure of NarGHI mutant NarG-R94S | 3IR7; http://dx.doi.org/10.2210/pdb3IR7/pdb | Publicly available at the RCSB Protein Data Bank. |
| Thoden JB, Wesenberg G, Raushel FM, Holden HM | 1998 | Structure of Carbamoyl Phosphate Synthetase Complexed with the ATP Analog AMPPNP | 1BXR; http://dx.doi.org/10.2210/pdb1BXR/pdb | Publicly available at the RCSB Protein Data Bank. |
| Schumacher MA | 2008 | MDT Protein | 3DNV; http://dx.doi.org/10.2210/pdb3DNV/pdb | Publicly available at the RCSB Protein Data Bank. |
| Stevens RC, Gouaux JE, Lipscomb WN | 1990 | Structural Consequences of Effector Binding to the T State of Aspartate Carbamoyltransferase. Crystal Structures of the Unligated and ATP-, and CTP-Complexed Enzymes AT 2.6-Angstroms Resolution | 5AT1; http://dx.doi.org/10.2210/pdb5AT1/pdb | Publicly available at the RCSB Protein Data Bank. |
| Schuwirth BS, Borovinskaya MA, Hau CW, Zhang W, Vila-Sanjurjo A, Holton JM, Cate JHD | 2005 | Crystal structure of the bacterial ribosome from Escherichia coli at 3.5 A resolution. This file contains the 30S subunit of one 70S ribosome. The entire crystal structure contains two 70S ribosomes and is described in remark 400 | 2AVY; http://dx.doi.org/10.2210/pdb2AVY/pdb | Publicly available at the RCSB Protein Data Bank. |
| Saikrishnan K, Wigley DB | 2009 | Crystal structure of the complete initiation complex of RecBCD | 3K70; http://dx.doi.org/10.2210/pdb3K70/pdb | Publicly available at the RCSB Protein Data Bank. |
| Miles BW, Thoden JB, Holden HM, Raushel FM | 2001 | Inactivation of the Amidotransferase Activity of Carbamoyl Phosphate Synthetase by the Antibiotic Acivicin | 1KEE; http://dx.doi.org/10.2210/pdb1KEE/pdb | Publicly available at the RCSB Protein Data Bank. |
| Wang S, Fleming RT, Westbrook EM, Matsumura P, McKay DB | 2005 | Structure of the Escherichia coli FlhDC complex, a prokaryotic heteromeric regulator of transcription | 2AVU; http://dx.doi.org/10.2210/pdb2AVU/pdb | Publicly available at the RCSB Protein Data Bank. |
| Safro M, Klipcan L, Moor N | 2011 | Crystal structure of Thermus thermophilus Phenylalanyl-tRNA synthetase comlexed with L-dopa | 3TEH; http://dx.doi.org/10.2210/pdb3TEH/pdb | Publicly available at the RCSB Protein Data Bank. |

| | | | | |
|---|---|---|---|---|
| Jin L, Stec B, Kantrowitz ER | 2000 | Crystal Structure of E. coli Aspartate Transcarbamoylase P268A Mutant in the R-State in the Presence of N-Phosphonacetyl-L-Aspartate | 1F1B; http://dx.doi.org/10.2210/pdb1F1B/pdb | Publicly available at the RCSB Protein Data Bank. |
| Oldham ML, Khare D, Quiocho FA, Davidson AL, Chen J | 2007 | The Crystal Structure of the E. coli Maltose Transporter | 2R6G; http://dx.doi.org/10.2210/pdb2R6G/pdb | Publicly available at the RCSB Protein Data Bank. |
| Wang J, Song JJ, Franklin MC, Kamtekar S, Im YJ, Rho SH, Seong IS, Lee CS, Chung CH, Eom SH | 2000 | Crystal Structures of the HSLVU Peptidase-Atpase Complex Reveal an ATP-Dependent Proteolysis Mechanism | 1G4A; http://dx.doi.org/10.2210/pdb1G4A/pdb | Publicly available at the RCSB Protein Data Bank. |
| Thoden J, Holden H | 1998 | Carbamoyl Phosphate Synthetase: Caught in the Act of Glutamine Hydrolysis | 1A9X; http://dx.doi.org/10.2210/pdb1A9X/pdb | Publicly available at the RCSB Protein Data Bank. |
| Schuwirth BS, Vila-Sanjurjo A, Cate JHD | 2006 | Crystal structure of the bacterial ribosome from escherichia coli in complex with the antibiotic kasugamyin at 3.5a resolution. this file contains the 30s subunit of one 70s ribosome. the entire crystal structure contains two 70s ribosomes and is described in remark 400 | 1VS7; http://dx.doi.org/10.2210/pdb1VS7/pdb | Publicly available at the RCSB Protein Data Bank. |
| Zhou J, Lancaster L, Trakhanov S, Noller HF | 2011 | Crystal Structure of Release Factor RF3 Trapped in the GTP State on a Rotated Conformation of the Ribosome (without viomycin) | 3UOQ; http://dx.doi.org/10.2210/pdb3UOQ/pdb | Publicly available at the RCSB Protein Data Bank. |
| Sharma A, Bonsor DA, Kleanthous C | 2009 | Crystal structure of processed TolB in complex with Pal | 2W8B; http://dx.doi.org/10.2210/pdb2W8B/pdb | Publicly available at the RCSB Protein Data Bank. |
| Chaudhry C, Horwich AL, Brunger AT, Adams PD | 2004 | GroEL-GroES-ADP7 | s1SX4; http://dx.doi.org/10.2210/pdb1SX4/pdb | Publicly available at the RCSB Protein Data Bank. |
| Honzatko RB, Crawford JL, Monaco HL, Ladner JE, Edwards BFP, Evans DR, Warren SG, Wiley DC, Ladner RC, Lipscomb WN | 1982 | Crystal and Molecular Structures of Native and CTP-Liganded Aspartate Carbamoyltransferase from Escherichia Coli | 2ATC; http://dx.doi.org/10.2210/pdb2ATC/pdb | Publicly available at the RCSB Protein Data Bank. |
| Zhou J, Lancaster L, Trakhanov S, Noller HF | 2011 | Crystal Structure of Release Factor RF3 Trapped in the GTP State on a Rotated Conformation of the Ribosome | 3SFS; http://dx.doi.org/10.2210/pdb3SFS/pdb | Publicly available at the RCSB Protein Data Bank. |
| Boal AK, Cotruvo JA Jnr, Stubbe J, Rosenzweig AC | 2010 | Ribonucleotide Reductase Dimanganese(II)-NrdF from Escherichia coli in Complex with NrdI | 3N39; http://dx.doi.org/10.2210/pdb3N39/pdb | Publicly available at the RCSB Protein Data Bank. |
| Heng S, Stieglitz KA, Eldo J, Xia J, Cardia JP, Kantrowitz ER | 2006 | The Structure of Wild-Type E. Coli Aspartate Transcarbamoylase in Complex with Novel T State Inhibitors at 2.25 Resolution | 2FZG; http://dx.doi.org/10.2210/pdb2FZG/pdb | Publicly available at the RCSB Protein Data Bank. |

| | | | | |
|---|---|---|---|---|
| Le Trong I, Aprikian P, Stenkamp RE, Sokurenko EV | 2009 | Complex of FimC, FimF, FimG and FimH | 3JWN; http://dx.doi.org/10.2210/pdb3JWN/pdb | Publicly available at the RCSB Protein Data Bank. |
| Shibata N | 2010 | Crystal structure of ethanolamine ammonia-lyase from Escherichia coli complexed with CN-CBL and (S)-2-amino-1-propanol | 3AO0; http://dx.doi.org/10.2210/pdb3AO0/pdb | Publicly available at the RCSB Protein Data Bank. |
| Schwieters CD, Suh JY, Grishaev A, Guirlando R, Takayama Y, Clore G.M | 2010 | Solution Structure of the Enzyme I Dimer Complexed with HPr Using Residual Dipolar Couplings and Small Angle X-Ray Scattering | 2XDF; http://dx.doi.org/10.2210/pdb2XDF/pdb | Publicly available at the RCSB Protein Data Bank. |
| Dunkle JA, Xiong L, Mankin AS, Cate JHD | 2010 | Crystal structure of the E. coli ribosome bound to chloramphenicol. This file contains the 30S subunit of the first 70S ribosome | 3OFA; http://dx.doi.org/10.2210/pdb3OFA/pdb | Publicly available at the RCSB Protein Data Bank. |
| Dunkle JA, Xiong L, Mankin AS, Cate JHD | 2010 | Crystal structure of the E. coli ribosome bound to chloramphenicol. This file contains the 30S subunit of the second 70S ribosome | 3OFB; http://dx.doi.org/10.2210/pdb3OFB/pdb | Publicly available at the RCSB Protein Data Bank. |
| Dunkle JA, Xiong L, Mankin AS, Cate JHD | 2010 | Crystal structure of the E. coli ribosome bound to chloramphenicol. This file contains the 50S subunit of the first 70S ribosome with chloramphenicol bound | 3OFC; http://dx.doi.org/10.2210/pdb3OFC/pdb | Publicly available at the RCSB Protein Data Bank. |
| Dunkle J.A, Xiong L, Mankin AS, Cate JHD | 2010 | Crystal structure of the E. coli ribosome bound to chloramphenicol. This file contains the 50S subunit of the second 70S ribosome | 3OFD; http://dx.doi.org/10.2210/pdb3OFD/pdb | Publicly available at the RCSB Protein Data Bank. |
| Hung CS, Bouckaert J | 2001 | FIMH adhesin-FIMC chaperone complex with D-mannose | 1KLF; http://dx.doi.org/10.2210/pdb1KLF/pdb | Publicly available at the RCSB Protein Data Bank. |
| Shibata N | 2010 | Crystal structure of ethanolamine ammonia-lyase from escherichia coli complexed with CN-CBL and (R)-2-amino-1-propanol | 3ANY; http://dx.doi.org/10.2210/pdb3ANY/pdb | Publicly available at the RCSB Protein Data Bank. |
| Dunkle JA, Xiong L, Mankin AS, Cate JHD | 2010 | Crystal structure of the E. coli ribosome bound to clindamycin. This file contains the 30S subunit of the first 70S ribosome | 3OFX; http://dx.doi.org/10.2210/pdb3OFX/pdb | Publicly available at the RCSB Protein Data Bank. |
| Dunkle JA, Xiong L, Mankin AS, Cate JHD | 2010 | Crystal structure of the E. coli ribosome bound to clindamycin. This file contains the 30S subunit of the second 70S ribosome | 3OFY; http://dx.doi.org/10.2210/pdb3OFY/pdb | Publicly available at the RCSB Protein Data Bank. |
| Dunkle JA, Xiong L, Mankin AS, Cate JHD | 2010 | Crystal structure of the E. coli ribosome bound to clindamycin. This file contains the 50S subunit of the first 70S ribosome bound to clindamycin | 3OFZ; http://dx.doi.org/10.2210/pdb3OFZ/pdb | Publicly available at the RCSB Protein Data Bank. |

| | | | | |
|---|---|---|---|---|
| Dunkle JA, Xiong L, Mankin AS, Cate JHD | 2010 | Crystal structure of the E. coli ribosome bound to erythromycin. This file contains the 30S subunit of the second 70S ribosome | 3OFP; http://dx.doi.org/10.2210/pdb3OFP/pdb | Publicly available at the RCSB Protein Data Bank. |
| Chinardet N, Welch M, Mourey L, Birck C, Samama JP | 1997 | Chey-binding domain of chea in complex with chey | 1A0O; http://dx.doi.org/10.2210/pdb1A0O/pdb | Publicly available at the RCSB Protein Data Bank. |
| Dunkle JA, Xiong L, Mankin AS, Cate JHD | 2010 | Crystal structure of the E. coli ribosome bound to erythromycin. This file contains the 50S subunit of the first 70S ribosome with erthromycin bound | 3OFR; http://dx.doi.org/10.2210/pdb3OFR/pdb | Publicly available at the RCSB Protein Data Bank. |
| Stieglitz KA, Dusinberre KJ, Cardia JP, Tsuruta H, Kantrowitz ER | 2005 | Structure of D236A mutant E. coli Aspartate Transcarbamoylase in presence of Phosphonoacetamide at 2.90 A resolution | 2A0F; http://dx.doi.org/10.2210/pdb2A0F/pdb | Publicly available at the RCSB Protein Data Bank. |
| Xu Z, Horwich AL, Sigler PB | 1997 | Crystal structure of the asymmetric chaperonin complex groel/groes/(ADP)7 | 1AON; http://dx.doi.org/10.2210/pdb1AON/pdb | Publicly available at the RCSB Protein Data Bank. |
| Borovinskaya MA, Shoji S, Holton JM, Fredrick K, Cate JHD | 2007 | Crystal structure of the bacterial ribosome from Escherichia coli in complex with spectinomycin and neomycin. This file contains the 30S subunit of the second 70S ribosome, with spectinomycin and neomycin bound. The entire crystal structure contains two 70S ribosomes | 2QP0; http://dx.doi.org/10.2210/pdb2QP0/pdb | Publicly available at the RCSB Protein Data Bank. |
| Clore GM, Wang G | 2000 | Complex of enzyme IIAGLC and the histidine-containing phosphocarrier protein HPR from Escherichia coli NMR, restrained regularized mean structure | 1GGR; http://dx.doi.org/10.2210/pdb1GGR/pdb | Publicly available at the RCSB Protein Data Bank. |
| Gouet P, Chinardet N, Welch M, Guillet V, Birck C, Mourey L, Samama JP | 2000 | Chey-binding domain of chea in complex with chey from crystals soaked in acetyl phosphate | 1FFS; http://dx.doi.org/10.2210/pdb1FFS/pdb | Publicly available at the RCSB Protein Data Bank. |
| Phan G, Remaut H, Lebedev A, Geibel S, Waksman G | 2011 | Crystal structure of the FimD usher bound to its cognate FimC: FimH substrate | 3RFZ; http://dx.doi.org/10.2210/pdb3RFZ/pdb | Publicly available at the RCSB Protein Data Bank. |
| Hung CS, Bouckaert J | 2001 | FimH adhesin Q133N mutant-FimC chaperone complex with methyl-alpha-D-mannose | 1KIU; http://dx.doi.org/10.2210/pdb1KIU/pdb | Publicly available at the RCSB Protein Data Bank. |
| Schuwirth B.S, Borovinskaya M.A, Hau C.W, Zhang W, Vila-Sanjurjo A, Holton J.M, Cate J.H.D | 2005 | Crystal structure of the bacterial ribosome from Escherichia coli at 3.5 A resolution. This file contains the 50S subunit of the second 70S ribosome. The entire crystal structure contains two 70S ribosomes and is described in remark 400 | 2AWB; http://dx.doi.org/10.2210/pdb2AWB/pdb | Publicly available at the RCSB Protein Data Bank. |

| | | | | |
|---|---|---|---|---|
| Gouaux J.E, Lipscomb W.N | 1989 | CRYSTAL STRUCTURES OF PHOSPHONOACETAMIDE LIGATED T AND PHOSPHONOACETAMIDE AND MALONATE LIGATED R STATES OF ASPARTATE CARBAMOYLTRANSFERASE AT 2.8-ANGSTROMS RESOLUTION AND NEUTRAL P*H | 1AT1; http://dx.doi.org/10.2210/pdb1AT1/pdb | Publicly available at the RCSB Protein Data Bank. |
| Oldham M.L, Chen J | 2010 | Crystal Structure of a pre-translocation state MBP-Maltose transporter complex bound to AMP-PNP | 3PUZ; http://dx.doi.org/10.2210/pdb3PUZ/pdb | Publicly available at the RCSB Protein Data Bank. |
| Oldham M.L, Chen J | 2010 | Crystal Structure of an outward-facing MBP-Maltose transporter complex bound to ADP-BeF3 | 3PUX; http://dx.doi.org/10.2210/pdb3PUX/pdb | Publicly available at the RCSB Protein Data Bank. |
| Oldham M.L, Chen J | 2010 | Crystal Structure of an outward-facing MBP-Maltose transporter complex bound to AMP-PNP after crystal soaking of the pretranslocation state | 3PUY; http://dx.doi.org/10.2210/pdb3PUY/pdb | Publicly available at the RCSB Protein Data Bank. |
| Oldham M.L, Chen J | 2010 | Crystal Structure of an outward-facing MBP-Maltose transporter complex bound to ADP-VO4 | 3PUV; http://dx.doi.org/10.2210/pdb3PUV/pdb | Publicly available at the RCSB Protein Data Bank. |
| Oldham M.L, Chen J | 2010 | Crystal Structure of an outward-facing MBP-Maltose transporter complex bound to ADP-AlF4 | 3PUW; http://dx.doi.org/10.2210/pdb3PUW/pdb | Publicly available at the RCSB Protein Data Bank. |
| Eldo J, Cardia J.P, O'Day E.M, Xia J, Tsuruta H, Kantrowitz E.R | 2006 | Structure of wild-type E. coli Aspartate Transcarbamoylase in the presence of N-phosphonacetyl-L-isoasparagine at 2.3A resolution | 2H3E; http://dx.doi.org/10.2210/pdb2H3E/pdb | Publicly available at the RCSB Protein Data Bank. |
| Dunkle J.A, Xiong L, Mankin A.S, Cate J.H.D | 2010 | Crystal structure of the E. coli ribosome bound to clindamycin. This file contains the 50S subunit of the second 70S ribosome | 3OG0; http://dx.doi.org/10.2210/pdb3OG0/pdb | Publicly available at the RCSB Protein Data Bank. |
| Stevens R.C, Gouaux J.E, Lipscomb W.N | 1990 | STRUCTURAL CONSEQUENCES OF EFFECTOR BINDING TO THE T STATE OF ASPARTATE CARBAMOYLTRANSFERASE. CRYSTAL STRUCTURES OF THE UNLIGATED AND ATP-, AND CTP-COMPLEXED ENZYMES AT 2.6-ANGSTROMS RESOLUTION | 4AT1; http://dx.doi.org/10.2210/pdb4AT1/pdb | Publicly available at the RCSB Protein Data Bank. |
| Borovinskaya M.A, Pai R.D, Zhang W, Schuwirth B.-S, Holton J.M, Hirokawa G, Kaji H, Kaji A, Cate J.H.D | 2007 | Crystal structure of the bacterial ribosome from Escherichia coli in complex with paromomycin and ribosome recycling factor (RRF). This file contains the 30S subunit of the second 70S ribosome, with paromomycin bound. The entire crystal structure contains two 70S ribosomes and is described in remark 400 | 2Z4M; http://dx.doi.org/10.2210/pdb2Z4M/pdb | Publicly available at the RCSB Protein Data Bank. |

| | | | | |
|---|---|---|---|---|
| Ruprecht J, Yankovskaya V, Maklashina E, Iwata S, Cecchini G | 2009 | E. coli succinate:quinone oxidoreductase (SQR) with carboxin bound | 2WDQ; http://dx.doi.org/10.2210/pdb2WDQ/pdb | Publicly available at the RCSB Protein Data Bank. |
| Thoden J.B, Holden H.M, Wesenberg G, Raushel F.M, Rayment I | 1997 | CARBAMOYL PHOSPHATE SYNTHETASE FROM ESCHERICHIA COLI | 1JDB; http://dx.doi.org/10.2210/pdb1JDB/pdb | Publicly available at the RCSB Protein Data Bank. |
| Ruprecht J, Yankovskaya V, Maklashina E, Iwata S, Cecchini G | 2009 | Crystal structure of the E. coli succinate:quinone oxidoreductase ( SQR) SdhB His207Thr mutant | 2WP9; http://dx.doi.org/10.2210/pdb2WP9/pdb | Publicly available at the RCSB Protein Data Bank. |
| Numata T, Fukai S, Ikeuchi Y, Suzuki T, Nureki O | 2005 | crystal structure of heterohexameric TusBCD proteins, which are crucial for the tRNA modification | 2D1P; http://dx.doi.org/10.2210/pdb2D1P/pdb | Publicly available at the RCSB Protein Data Bank. |
| Zhang W, Dunkle J.A, Cate J.H.D | 2009 | Crystal structure of the E. coli 70S ribosome in an intermediate state of ratcheting | 3I1O; http://dx.doi.org/10.2210/pdb3I1O/pdb | Publicly available at the RCSB Protein Data Bank. |
| Zhang W, Dunkle J.A, Cate J.H.D | 2009 | Crystal structure of the E. coli 70S ribosome in an intermediate state of ratcheting | 3I1N; http://dx.doi.org/10.2210/pdb3I1N/pdb | Publicly available at the RCSB Protein Data Bank. |
| Zhang W, Dunkle J.A, Cate J.H.D | 2009 | Crystal structure of the E. coli 70S ribosome in an intermediate state of ratcheting | 3I1M; http://dx.doi.org/10.2210/pdb3I1M/pdb | Publicly available at the RCSB Protein Data Bank. |
| Bertero M.G, Rothery R.A, Weiner J.H, Strynadka N.C.J | 2009 | Crystal structure of NarGHI mutant NarG-H49C | 3IR5; http://dx.doi.org/10.2210/pdb3IR5/pdb | Publicly available at the RCSB Protein Data Bank. |
| Khare D, Oldham M.L, Orelle C, Davidson A.L, Chen J | 2008 | Crystal structure of the resting state maltose transporter from E. coli | 3FH6; http://dx.doi.org/10.2210/pdb3FH6/pdb | Publicly available at the RCSB Protein Data Bank. |
| Sang Y.S, Cai M, Clore G.M | 2009 | NMR structure of the IIAchitobiose-IIBchitobiose phosphoryl transition state complex of the N,N'-diacetylchitoboise brance of the E. coli phosphotransferase system | 2WY2; http://dx.doi.org/10.2210/pdb2WY2/pdb | Publicly available at the RCSB Protein Data Bank. |
| Stieglitz K, Stec B, Baker D.P, Kantrowitz E.R | 2004 | Aspartate Transcarbamoylase Catalytic Chain Mutant E50A Complex with Phosphonoacetamide, Malonate, and Cytidine-5-Prime-Triphosphate (CTP) | 1TUG; http://dx.doi.org/10.2210/pdb1TUG/pdb | Publicly available at the RCSB Protein Data Bank. |
| Zhang W, Dunkle J.A, Cate J.H.D | 2009 | Crystal structure of the E. coli 70S ribosome in an intermediate state of ratcheting | 3I1S; http://dx.doi.org/10.2210/pdb3I1S/pdb | Publicly available at the RCSB Protein Data Bank. |
| Iverson T.M, Luna-Chavez C, Croal L.R, Cecchini G, Rees D.C | 2002 | Quinol-Fumarate Reductase with Menaquinol Molecules | 1L0V; http://dx.doi.org/10.2210/pdb1L0V/pdb | Publicly available at the RCSB Protein Data Bank. |
| Zhang W, Dunkle J.A, Cate J.H.D | 2009 | Crystal structure of the E. coli 70S ribosome in an intermediate state of ratcheting | 3I1Q; http://dx.doi.org/10.2210/pdb3I1Q/pdb | Publicly available at the RCSB Protein Data Bank. |
| Ishikawa M, Tsuchiya D, Oyama T, Tsunaka Y, Morikawa K | 2004 | fatty acid beta-oxidation multienzyme complex from Pseudomonas fragi, form I (native2) | 1WDK; http://dx.doi.org/10.2210/pdb1WDK/pdb | Publicly available at the RCSB Protein Data Bank. |

| | | | | |
|---|---|---|---|---|
| Iverson T.M, Luna-Chavez C, Croal L.R, Cecchini G, Rees D.C | 2001 | E. coli Quinol-Fumarate Reductase with Bound Inhibitor HQNO | 1KF6; http://dx.doi.org/10.2210/pdb1KF6/pdb | Publicly available at the RCSB Protein Data Bank. |
| Zhang W, Dunkle J.A, Cate J.H.D | 2009 | Crystal structure of the E. coli 70S ribosome in an intermediate state of ratcheting | 3I1Z; http://dx.doi.org/10.2210/pdb3I1Z/pdb | Publicly available at the RCSB Protein Data Bank. |
| Daniels J.N, Schindelin H | 2007 | Crystal Structure of Activated MPT Synthase | 3BII; http://dx.doi.org/10.2210/pdb3BII/pdb | Publicly available at the RCSB Protein Data Bank. |
| Jormakka M, Richardson D, Byrne B, Iwata S | 2003 | Crystal Structure of NarGH complex | 1R27; http://dx.doi.org/10.2210/pdb1R27/pdb | Publicly available at the RCSB Protein Data Bank. |
| Borovinskaya M.A, Shoji S, Holton J.M, Fredrick K, Cate J.H.D | 2007 | Crystal structure of the bacterial ribosome from Escherichia coli in complex with spectinomycin. This file contains the 30S subunit of the first 70S ribosome, with spectinomycin bound. The entire crystal structure contains two 70S ribosomes | 2QOU; http://dx.doi.org/10.2210/pdb2QOU/pdb | Publicly available at the RCSB Protein Data Bank. |
| Gouaux J.E, Lipscomb W.N | 1989 | CRYSTAL STRUCTURES OF PHOSPHONOACETAMIDE LIGATED T AND PHOSPHONOACETAMIDE AND MALONATE LIGATED R STATES OF ASPARTATE CARBAMOYLTRANSFERASE AT 2.8-ANGSTROMS RESOLUTION AND NEUTRAL PH | 2AT1; http://dx.doi.org/10.2210/pdb2AT1/pdb | Publicly available at the RCSB Protein Data Bank. |
| Gouet P, Chinardet N, Welch M, Guillet V, Birck C, Mourey L, Samama J.-P | 2000 | CHEY-BINDING DOMAIN OF CHEA IN COMPLEX WITH CHEY AT 2.1 A RESOLUTION | 1FFG; http://dx.doi.org/10.2210/pdb1FFG/pdb | Publicly available at the RCSB Protein Data Bank. |
| Dunkle J.A, Wang L, Feldman M.B, Pulk A, Chen V.B, Kapral G.J, Noeske J, Richardson J.S, Blanchard S.C, Cate J.H.D | 2011 | Structures of the bacterial ribosome in classical and hybrid states of tRNA binding | 3R8S; http://dx.doi.org/10.2210/pdb3R8S/pdb | Publicly available at the RCSB Protein Data Bank. |
| Ruprecht J, Yankovskaya V, Maklashina E, Iwata S, Cecchini G | 2009 | E. coli succinate:quinone oxidoreductase (SQR) with an empty quinone- binding pocket | 2WDV; http://dx.doi.org/10.2210/pdb2WDV/pdb | Publicly available at the RCSB Protein Data Bank. |
| Cardia J.P, Eldo J, Xia J, O'Day E.M, Tsuruta H, Kantrowitz E.R | 2006 | E. coli Aspartate Transcarbamoylase complexed with N-phosphonacetyl-L-asparagine | 2IPO; http://dx.doi.org/10.2210/pdb2IPO/pdb | Publicly available at the RCSB Protein Data Bank. |
| Baradaran R, Berrisford J.M, Minhas G.S, Sazanov L.A | 2012 | Crystal structure of the entire respiratory complex I from Thermus thermophilus | 4HEA; http://dx.doi.org/10.2210/pdb4HEA/pdb | Publicly available at the RCSB Protein Data Bank. |
| Bingel-Erlenmeyer R, Kohler R, Kramer G, Sandikci A, Antolic S, Maier T, Schaffitzel C, Wiedmann B, Bukau B, Ban N | 2007 | Structure of PDF binding helix in complex with the ribosome | 2VHM; http://dx.doi.org/10.2210/pdb2VHM/pdb | Publicly available at the RCSB Protein Data Bank. |
| Choudhury D, Thompson A, Stojanoff V, Langerman S, Pinkner J, Hultgren S.J, Knight S | 1999 | X-RAY STRUCTURE OF THE FIMC-FIMH CHAPERONE ADHESIN COMPLEX FROM UROPATHOGENIC E.COLI | 1QUN; http://dx.doi.org/10.2210/pdb1QUN/pdb | Publicly available at the RCSB Protein Data Bank. |

| Stec B, Williams M.K, Stieglitz K.A, Kantrowitz E.R | 2007 | Structure of regulatory chain mutant H20A of asparate transcarbamoylase from E. coli | 2QGF; http://dx.doi.org/10.2210/pdb2QGF/pdb | Publicly available at the RCSB Protein Data Bank. |
|---|---|---|---|---|
| Rothery R.A, Bertero M.G, Cammack R, Palak M, Blasco F, Strynadka N.C, Weiner J.H | 2004 | Crystal structure of the apomolybdo-NarGHI | 1SIW; http://dx.doi.org/10.2210/pdb1SIW/pdb | Publicly available at the RCSB Protein Data Bank. |
| Yankovskaya V, Horsefield R, Tornroth S, Luna-Chavez C, Miyoshi H, Leger C, Byrne B, Cecchini G, Iwata S | 2002 | Complex II (Succinate Dehydrogenase) From E. Coli with ubiquinone bound | 1NEK; http://dx.doi.org/10.2210/pdb1NEK/pdb | Publicly available at the RCSB Protein Data Bank. |
| Abramson J, Riistama S, Larsson G, Jasaitis A, Svensson-Ek M, Puustinen A, Iwata S, Wikstrom M | 2000 | The structure of ubiquinol oxidase from Escherichia E. coli | 1FFT; http://dx.doi.org/10.2210/pdb1FFT/pdb | Publicly available at the RCSB Protein Data Bank. |
| Yankovskaya V, Horsefield R, Tornroth S, Luna-Chavez C, Miyoshi H, Leger C, Byrne B, Cecchini G, Iwata S | 2002 | Complex II (Succinate Dehydrogenase) From E. Coli with Dinitrophenol-17 inhibitor co-crystallized at the ubiquinone binding site | 1NEN; http://dx.doi.org/10.2210/pdb1NEN/pdb | Publicly available at the RCSB Protein Data Bank. |
| Xia D, Maurizi M.R, Guo F, Singh S.K, Esser L | 2003 | ClpNS with fragments | 1R6Q; http://dx.doi.org/10.2210/pdb1R6Q/pdb | Publicly available at the RCSB Protein Data Bank. |
| Ruprecht J, Yankovskaya V, Maklashina E, Iwata S, Cecchini G | 2009 | E. coli succinate:quinone oxidoreductase (SQR) with pentachlorophenol bound | 2WDR; http://dx.doi.org/10.2210/pdb2WDR/pdb | Publicly available at the RCSB Protein Data Bank. |
| Stec B, Williams M.K, Stieglitz K.A, Kantrowitz E.R | 2007 | Structure of a regulatory subunit mutant D19A of ATCase from E. coli | 2QG9; http://dx.doi.org/10.2210/pdb2QG9/pdb | Publicly available at the RCSB Protein Data Bank. |
| Schumacher M.A | 2009 | Structure of mdt protein | 3HZI; http://dx.doi.org/10.2210/pdb3HZI/pdb | Publicly available at the RCSB Protein Data Bank. |
| Johnson E, Nguyen P, Rees D.C | 2011 | Inward facing conformations of the MetNI methionine ABC transporter: CY5 SeMet soak crystal form | 3TUZ; http://dx.doi.org/10.2210/pdb3TUZ/pdb | Publicly available at the RCSB Protein Data Bank. |
| Manjasetty A.B, Powlowski J, Vrielink A | 2003 | Crystal structure of a bifunctional aldolase-dehydrogenase: sequestering a reactive and volatile intermediate | 1NVM; http://dx.doi.org/10.2210/pdb1NVM/pdb | Publicly available at the RCSB Protein Data Bank. |
| Cottevieille M, Larquet E, Jonic S, Petoukhov M.V, Caprini G, Paravisi S, Svergun D.I, Vanoni M.A, Boisset N | 2007 | The subnanometer cryo-electron microscopy structure of the nitrogen assimilating enzyme glutamate synthase reveals a 1.2 MDa heterohexameric complex | 2VDC; http://dx.doi.org/10.2210/pdb2VDC/pdb | Publicly available at the RCSB Protein Data Bank. |
| Huang J, Lipscomb W.N | 2003 | Aspartate Transcarbamylase (ATCase) of Escherichia coli: A New Crystalline R State Bound to PALA, or to Product Analogues Phosphate and Citrate | 1Q95; http://dx.doi.org/10.2210/pdb1Q95/pdb | Publicly available at the RCSB Protein Data Bank. |
| Rudolph M.J, Wuebbens M.M, Turque O, Rajagopalan K.V, Schindelin H | 2003 | Orthorhombic Crystal Form of Molybdopterin Synthase | 1NVI; http://dx.doi.org/10.2210/pdb1NVI/pdb | Publicly available at the RCSB Protein Data Bank. |

| Bingel-Erlenmeyer R, Kohler R, Kramer G, Sandikci A, Antolic S, Maier T, Schaffitzel C, Wiedmann B, Bukau B, Ban N | 2007 | Structure of PDF binding helix in complex with the ribosome. (Structure 2 of 4) | 2VHN; http://dx.doi.org/10.2210/pdb2VHN/pdb | Publicly available at the RCSB Protein Data Bank. |
|---|---|---|---|---|
| Borovinskaya M.A, Pai R.D, Zhang W, Schuwirth B.-S, Holton J.M, Hirokawa G, Kaji H, Kaji A, Cate J.H.D | 2007 | Crystal structure of the bacterial ribosome from Escherichia coli in complex with gentamicin. This file contains the 30S subunit of the first 70S ribosome, with gentamicin bound. The entire crystal structure contains two 70S ribosomes and is described in remark 400 | 2QB9; http://dx.doi.org/10.2210/pdb2QB9/pdb | Publicly available at the RCSB Protein Data Bank. |
| Bertero M.G, Strynadka N.C.J | 2003 | Crystal structure of Nitrate Reductase A NarGHI, from Escherichia coli | 1Q16; http://dx.doi.org/10.2210/pdb1Q16/pdb | Publicly available at the RCSB Protein Data Bank. |
| Johnson E, Nguyen P, Rees D.C | 2011 | Inward facing conformations of the MetNI methionine ABC transporter: DM crystal form | 3TUJ; http://dx.doi.org/10.2210/pdb3TUJ/pdb | Publicly available at the RCSB Protein Data Bank. |
| Berk V, Zhang W, Pai R.D, Cate J.H.D | 2006 | Crystal Structure of Ribosome with messenger RNA and the Anticodon stem-loop of P-site tRNA. This file contains the 30s subunit of one 70s ribosome. The entire crystal structure contains two 70s ribosomes and is described in remark 400 | 2I2P; http://dx.doi.org/10.2210/pdb2I2P/pdb | Publicly available at the RCSB Protein Data Bank. |
| Wang J, Kantrowitz E.R | 2006 | Structure of D236A E. coli Aspartate Transcarbamoylase in the presence of phosphonoacetamide and l-Aspartate at 2.60 A resolution | 2HSE; http://dx.doi.org/10.2210/pdb2HSE/pdb | Publicly available at the RCSB Protein Data Bank. |
| Johansson T, Pedersen A, Leckner J, Karlsson B.G | 2005 | Complex of the domain I and domain III of Escherichia coli transhydrogenase | 2BRU; http://dx.doi.org/10.2210/pdb2BRU/pdb | Publicly available at the RCSB Protein Data Bank. |
| Berk V, Zhang W, Pai R.D, Cate J.H.D | 2006 | Crystal Structure of Ribosome with messenger RNA and the Anticodon stem-loop of P-site tRNA. This file contains the 30s subunit of one 70s ribosome. The entire crystal structure contains two 70s ribosomes and is described in remark 400 | 2I2U; http://dx.doi.org/10.2210/pdb2I2U/pdb | Publicly available at the RCSB Protein Data Bank. |
| Thoden J.B, Huang X, Raushel F.M, Holden H.M | 1999 | Crystal structure of carbamoyl phosphate synthetase complexed at CYS269 in the small subunit with the tetrahedral mimic l-glutamate gamma-semialdehyde | 1CS0; http://dx.doi.org/10.2210/pdb1CS0/pdb | Publicly available at the RCSB Protein Data Bank. |
| Berk V, Zhang W, Pai R.D, Cate J.H.D | 2006 | Crystal Structure of Ribosome with messenger RNA and the Anticodon stem-loop of P-site tRNA. This file contains the 50s subunit of one 70s ribosome. The entire crystal structure contains two 70s ribosomes and is described in remark 400 | 2I2V; http://dx.doi.org/10.2210/pdb2I2V/pdb | Publicly available at the RCSB Protein Data Bank. |

| | | | | |
|---|---|---|---|---|
| Mougous J.D, Lee D.H, Hubbard S.C, Schelle M.W, Vocadlo D.J, Berger J.M, Bertozzi C.R | 2005 | Crystal Structure of a GTP-Regulated ATP Sulfurylase Heterodimer from Pseudomonas syringae | 1ZUN; http://dx.doi.org/10.2210/pdb1ZUN/pdb | Publicly available at the RCSB Protein Data Bank. |
| Jin L, Stec B, Lipscomb W.N, Kantrowitz E.R | 1999 | ASPARTATE TRANSCARBAMOYLASE COMPLEXED WITH N-PHOSPHONACETYL-L-ASPARTATE (PALA) | 1D09; http://dx.doi.org/10.2210/pdb1D09/pdb | Publicly available at the RCSB Protein Data Bank. |
| Gruswitz F, O'Connell J III, Stroud R.M, Center for Structures of Membrane Proteins (CSMP) | 2006 | Crystal structure of the E. coli ammonia channel AMTB complexed with the signal transduction protein GLNK | 2NS1; http://dx.doi.org/10.2210/pdb2NS1/pdb | Publicly available at the RCSB Protein Data Bank. |
| Boal A.K, Cotruvo J.A Jr, Stubbe J, Rosenzweig A.C | 2010 | Ribonucleotide Reductase Dimanganese(II)-NrdF from Escherichia coli in Complex with Reduced NrdI | 3N3A; http://dx.doi.org/10.2210/pdb3N3A/pdb | Publicly available at the RCSB Protein Data Bank. |
| Marinoni E.N, de Oliveira J.S, Nicolet Y, Raulfs E.C, Amara P, Dean D.R, Fontecilla-Camps J.C | 2012 | A. fulgidus IscS-IscU complex structure | 4EB7; http://dx.doi.org/10.2210/pdb4EB7/pdb | Publicly available at the RCSB Protein Data Bank. |
| Dunkle J.A, Zhang W, Cate J.H.D, Mankin A.S | 2010 | Crystal structure of the E. coli ribosome bound to CEM-101. This file contains the 50S subunit of the second 70S ribosome | 1VT2; http://dx.doi.org/10.2210/pdb1VT2/pdb | Publicly available at the RCSB Protein Data Bank. |
| Nishiyama M, Horst R, Eidam O, Herrmann T, Ignatov O, Vetsch M, Bettendorff P, Jelesarov I, Grutter M.G, Wuthrich K, Glockshuber R, Capitani G | 2005 | Crystal Structure of the Ternary Complex of FIMD (N-Terminal Domain) with FIMC and the Pilin Domain of FIMH | 1ZE3; http://dx.doi.org/10.2210/pdb1ZE3/pdb | Publicly available at the RCSB Protein Data Bank. |
| Alam N, Stieglitz K.A, Caban M.D, Gourinath S, Tsuruta H, Kantrowitz E.R | 2004 | E. coli Aspartate Transcarbamylase 240's Loop Mutant (K244N) | 1SKU; http://dx.doi.org/10.2210/pdb1SKU/pdb | Publicly available at the RCSB Protein Data Bank. |
| Huang J, Lipscomb W.N | 2005 | T-state Active Site of Aspartate Transcarbamylase: Crystal Structure of the Carbamyl Phosphate and L-alanosine Ligated Enzyme | 2AIR; http://dx.doi.org/10.2210/pdb2AIR/pdb | Publicly available at the RCSB Protein Data Bank. |
| Iverson T.M, Luna-Chavez C, Croal L.R, Cecchini G, Rees D.C | 2001 | QUINOL-FUMARATE REDUCTASE WITH QUINOL INHIBITOR 2-[1-(4-CHLORO-PHENYL)-ETHYL]-4,6-DINITRO-PHENOL | 1KFY; http://dx.doi.org/10.2210/pdb1KFY/pdb | Publicly available at the RCSB Protein Data Bank. |
| Minailiuc O.M, Ekiel I, Milad M, Montreal-Kingston Bacterial Structural Genomics Initiative (BSGI) | 2007 | NMR solution structure of NapD in complex with NapA1-35 signal peptide | 2PQ4; http://dx.doi.org/10.2210/pdb2PQ4/pdb | Publicly available at the RCSB Protein Data Bank. |
| Macol C.P, Tsuruta H, Stec B, Kantrowitz E.R | 2001 | CRYSTAL STRUCTURE OF MUTANT. R105A OF E. COLI ASPARTATE TRANSCARBAMOYLASE | 1I5O; http://dx.doi.org/10.2210/pdb1I5O/pdb | Publicly available at the RCSB Protein Data Bank. |
| Bertero M.G, Rothery R.A, Weiner J.H, Strynadka N.C.J | 2008 | The crystal structure of the NarGHI mutant NarH C16A | 3EGW; http://dx.doi.org/10.2210/pdb3EGW/pdb | Publicly available at the RCSB Protein Data Bank. |
| Li G, Zhang YInouye M, Ikura M | 2008 | Structure of E. coli toxin RelE (R81A/R83A) mutant in complex with antitoxin RelBc (K47–L79) peptide | 2KC8; http://dx.doi.org/10.2210/pdb2KC8/pdb | Publicly available at the RCSB Protein Data Bank. |

| | | | | |
|---|---|---|---|---|
| Conroy M.J, Durand A, Lupo D, Li X.-D, Bullough P.A, Winkler F.K, Merrick M | 2006 | Regulating the Escherichia coli ammonia channel: the crystal structure of the AmtB-GlnK complex | 2NUU; http://dx.doi.org/10.2210/pdb2NUU/pdb | Publicly available at the RCSB Protein Data Bank. |
| Zhou J, Lancaster L, Trakhanov S, Noller H.F | 2011 | Crystal Structure of Release Factor RF3 Trapped in the GTP State on a Rotated Conformation of the Ribosome | 3SGF; http://dx.doi.org/10.2210/pdb3SGF/pdb | Publicly available at the RCSB Protein Data Bank. |
| Dunkle J.A, Zhang W, Cate J.H.D, Mankin A.S | 2010 | Crystal structure of the E. coli ribosome bound to CEM-101. This file contains the 30S subunit of the first 70S ribosome | 3OR9; http://dx.doi.org/10.2210/pdb3OR9/pdb | Publicly available at the RCSB Protein Data Bank. |
| Dunkle J.A, Xiong L, Mankin A.S, Cate J.H.D | 2010 | Crystal structure of the E. coli ribosome bound to erythromycin. This file contains the 50S subunit of the second 70S ribosome | 3OFQ; http://dx.doi.org/10.2210/pdb3OFQ/pdb | Publicly available at the RCSB Protein Data Bank. |
| Eidam O, Grutter M.G, Capitani G | 2008 | Crystal structure of the ternary complex of FimD (N-Terminal Domain, FimDN) with FimC and the N-terminally truncated pilus subunit FimF (FimFt) | 3BWU; http://dx.doi.org/10.2210/pdb3BWU/pdb | Publicly available at the RCSB Protein Data Bank. |
| Cingolani G, Duncan T.M | 2010 | Structure of the E.coli F1-ATP synthase inhibited by subunit Epsilon | 3OAA; http://dx.doi.org/10.2210/pdb3OAA/pdb | Publicly available at the RCSB Protein Data Bank. |
| Dunkle J.A, Wang L, Feldman M.B, Pulk A, Chen V.B, Kapral G.J, Noeske J, Richardson J.S, Blanchard S.C, Cate J.H.D | 2011 | Structures of the bacterial ribosome in classical and hybrid states of tRNA binding | 3R8T; http://dx.doi.org/10.2210/pdb3R8T/pdb | Publicly available at the RCSB Protein Data Bank. |
| Dunkle J.A, Zhang W, Cate J.H.D, Mankin A.S | 2010 | Crystal structure of the E. coli ribosome bound to CEM-101. This file contains the 50S subunit of the first 70S ribosome bound to CEM-101 | 3ORB; http://dx.doi.org/10.2210/pdb3ORB/pdb | Publicly available at the RCSB Protein Data Bank. |
| Gouaux J.E, Stevens R.C, Lipscomb W.N | 1989 | CRYSTAL STRUCTURES OF ASPARTATE CARBAMOYLTRANSFERASE LIGATED WITH PHOSPHONOACETAMIDE, MALONATE, AND CTP OR ATP AT 2.8-ANGSTROMS RESOLUTION AND NEUTRAL P*H | 7AT1; http://dx.doi.org/10.2210/pdb7AT1/pdb | Publicly available at the RCSB Protein Data Bank. |
| Bertero M.G, Rothery R.A, Boroumand N, Palak M, Blasco F, Ginet N, Weiner J.H, Strynadka N.C.J | 2004 | The crystal structure of Nitrate Reductase A, NarGHI, in complex with the Q-site inhibitor pentachlorophenol | 1Y4Z; http://dx.doi.org/10.2210/pdb1Y4Z/pdb | Publicly available at the RCSB Protein Data Bank. |
| Zhang W, Dunkle J.A, Cate J.H.D | 2009 | Crystal structure of the E. coli 70S ribosome in an intermediate state of ratcheting | 3I22; http://dx.doi.org/10.2210/pdb3I22/pdb | Publicly available at the RCSB Protein Data Bank. |
| Rudolph M.J, Wuebbens M.M, Rajagolpalan K.V, Schindelin H | 2000 | MOLYBDOPTERIN SYNTHASE (MOAD/MOAE) | 1FM0; http://dx.doi.org/10.2210/pdb1FM0/pdb | Publicly available at the RCSB Protein Data Bank. |
| Johnson E, Nguyen P.T, Rees D.C | 2011 | Inward facing conformations of the MetNI methionine ABC transporter: CY5 native crystal form | 3TUI; http://dx.doi.org/10.2210/pdb3TUI/pdb | Publicly available at the RCSB Protein Data Bank. |

| | | | | |
|---|---|---|---|---|
| Heng S, Stieglitz K.A, Eldo J, Xia J, Cardia J.P, Kantrowitz E.R | 2006 | The Structure of Wild-Type E. Coli Aspartate Transcarbamoylase in Complex with Novel T State Inhibitors at 2.50 Resolution | 2FZK; http://dx.doi.org/10.2210/pdb2FZK/pdb | Publicly available at the RCSB Protein Data Bank. |
| Zeth K, Ravelli R.B, Paal K, Cusack S, Bukau B, Dougan D.A | 2002 | The structural basis of ClpS-mediated switch in ClpA substrate recognition | 1MG9; http://dx.doi.org/10.2210/pdb1MG9/pdb | Publicly available at the RCSB Protein Data Bank. |
| Dunkle J.A, Wang L, Feldman M.B, Pulk A, Chen V.B, Kapral G.J, Noeske J, Richardson J.S, Blanchard S.C, Cate J.H.D | 2012 | Structures of the bacterial ribosome in classical and hybrid states of tRNA binding | 4GD2; http://dx.doi.org/10.2210/pdb4GD2/pdb | Publicly available at the RCSB Protein Data Bank. |
| Sang Y.S, Cai M, Clore G.M | 2009 | NMR structure of the IIAchitobiose-IIBchitobiose complex of the N, N'- diacetylchitoboise brance of the E. coli phosphotransferase system | 2WWV; http://dx.doi.org/10.2210/pdb2WWV/pdb | Publicly available at the RCSB Protein Data Bank. |
| Bertero M.G, Rothery R.A, Weiner J.H, Strynadka N.C.J | 2009 | Crystal structure of NarGHI mutant NarG-H49S | 3IR6; http://dx.doi.org/10.2210/pdb3IR6/pdb | Publicly available at the RCSB Protein Data Bank. |
| Zhang W, Dunkle J.A, Cate J.H.D | 2009 | Crystal structure of the E. coli 70S ribosome in an intermediate state of ratcheting | 3I20; http://dx.doi.org/10.2210/pdb3I20/pdb | Publicly available at the RCSB Protein Data Bank. |
| Zhang W, Dunkle J.A, Cate J.H.D | 2009 | Crystal structure of the E. coli 70S ribosome in an intermediate state of ratcheting | 3I21; http://dx.doi.org/10.2210/pdb3I21/pdb | Publicly available at the RCSB Protein Data Bank. |
| Borovinskaya M.A, Pai R.D, Zhang W, Schuwirth B.-S, Holton J.M, Hirokawa G, Kaji H, Kaji A, Cate J.H.D | 2007 | Crystal structure of the bacterial ribosome from Escherichia coli in complex with neomycin. This file contains the 30S subunit of the first 70S ribosome, with neomycin bound. The entire crystal structure contains two 70S ribosomes and is described in remark 400 | 2QAL; http://dx.doi.org/10.2210/pdb2QAL/pdb | Publicly available at the RCSB Protein Data Bank. |
| Borovinskaya M.A, Pai R.D, Zhang W, Schuwirth B.-S, Holton J.M, Hirokawa G, Kaji H, Kaji A, Cate J.H.D | 2007 | Crystal structure of the bacterial ribosome from Escherichia coli in complex with neomycin. This file contains the 50S subunit of the first 70S ribosome, with neomycin bound. The entire crystal structure contains two 70S ribosomes and is described in remark 400 | 2QAM; http://dx.doi.org/10.2210/pdb2QAM/pdb | Publicly available at the RCSB Protein Data Bank. |
| Borovinskaya M.A, Pai R.D, Zhang W, Schuwirth B.-S, Holton J.M, Hirokawa G, Kaji H, Kaji A, Cate J.H.D | 2007 | Crystal structure of the bacterial ribosome from Escherichia coli in complex with neomycin. This file contains the 30S subunit of the second 70S ribosome, with neomycin bound. The entire crystal structure contains two 70S ribosomes and is described in remark 400 | 2QAN; http://dx.doi.org/10.2210/pdb2QAN/pdb | Publicly available at the RCSB Protein Data Bank. |

| | | | | |
|---|---|---|---|---|
| Stieglitz K.A, Xia J, Kantrowitz E.R | 2008 | Crystal structure of Wild-Type E. Coli Asparate Transcarbamoylase at pH 8.5 at 2.80 A Resolution | 3D7S; http://dx.doi.org/10.2210/pdb3D7S/pdb | Publicly available at the RCSB Protein Data Bank. |
| Borovinskaya M.A, Pai R.D, Zhang W, Schuwirth B.-S, Holton J.M, Hirokawa G, Kaji H, Kaji A, Cate J.H.D | 2007 | Crystal structure of the bacterial ribosome from Escherichia coli in complex with paromomycin and ribosome recycling factor (RRF). This file contains the 50S subunit of the first 70S ribosome, with paromomycin and RRF bound. The entire crystal structure contains two 70S ribosomes and is described in remark 400 | 2Z4L; http://dx.doi.org/10.2210/pdb2Z4L/pdb | Publicly available at the RCSB Protein Data Bank. |
| Ha Y, Allewell N.M | 1998 | ATCASE Y165F MUTANT | 9ATC; http://dx.doi.org/10.2210/pdb9ATC/pdb | Publicly available at the RCSB Protein Data Bank. |
| Borovinskaya M.A, Pai R.D, Zhang W, Schuwirth B.-S, Holton J.M, Hirokawa G, Kaji H, Kaji A, Cate J.H.D | 2007 | Crystal structure of the bacterial ribosome from Escherichia coli in complex with paromomycin and ribosome recycling factor (RRF). This file contains the 50S subunit of the second 70S ribosome, with paromomycin and RRF bound. The entire crystal structure contains two 70S ribosomes and is described in remark 400 | 2Z4N; http://dx.doi.org/10.2210/pdb2Z4N/pdb | Publicly available at the RCSB Protein Data Bank. |
| Borovinskaya M.A, Shoji S, Holton J.M, Fredrick K, Cate J.H.D | 2007 | Crystal structure of the bacterial ribosome from Escherichia coli in complex with spectinomycin and neomycin. This file contains the 50S subunit of the first 70S ribosome, with neomycin bound. The entire crystal structure contains two 70S ribosomes | 2QOZ; http://dx.doi.org/10.2210/pdb2QOZ/pdb | Publicly available at the RCSB Protein Data Bank. |
| Borovinskaya M.A, Pai R.D, Zhang W, Schuwirth B.-S, Holton J.M, Hirokawa G, Kaji H, Kaji A, Cate J.H.D | 2007 | Crystal structure of the bacterial ribosome from Escherichia coli in complex with paromomycin and ribosome recycling factor (RRF). This file contains the 30S subunit of the first 70S ribosome, with paromomycin bound. The entire crystal structure contains two 70S ribosomes and is described in remark 400 | 2Z4K; http://dx.doi.org/10.2210/pdb2Z4K/pdb | Publicly available at the RCSB Protein Data Bank. |
| Stevens R.C, Gouaux J.E, Lipscomb W.N | 1990 | Structural consequences of effector binding to the t state of aspartate carbamoyltransferase. Crystal structures of the unligated and atp-, and ctp-complexed enzymes at 2.6-angstroms resolution | 6AT1; http://dx.doi.org/10.2210/pdb6AT1/pdb | Publicly available at the RCSB Protein Data Bank. |
| Lovell S, Battaile K.P, Park K.-T, Wu W, Holyoak T, Lutkenhaus J | 2011 | 4.3A resolution structure of a MinD-MinE(I24N) protein complex | 3R9J; http://dx.doi.org/10.2210/pdb3R9J/pdb | Publicly available at the RCSB Protein Data Bank. |

| | | | | |
|---|---|---|---|---|
| Shibata N | 2009 | Crystal structure of ethanolamine ammonia-lyase from Escherichia coli complexed with CN-Cbl (substrate-free form) | 3ABR; http://dx.doi.org/10.2210/pdb3ABR/pdb | Publicly available at the RCSB Protein Data Bank. |
| Shibata N | 2009 | Crystal structure of ethanolamine ammonia-lyase from Escherichia coli complexed with adeninylpentylcobalamin and ethanolamine | 3ABS; http://dx.doi.org/10.2210/pdb3ABS/pdb | Publicly available at the RCSB Protein Data Bank. |
| Dunkle J.A, Wang L, Feldman M.B, Pulk A, Chen V.B, Kapral G.J, Noeske J, Richardson J.S, Blanchard S.C, Cate J.H.D | 2012 | Structures of the bacterial ribosome in classical and hybrid states of tRNA binding | 4GD1; http://dx.doi.org/10.2210/pdb4GD1/pdb | Publicly available at the RCSB Protein Data Bank. |
| Chaudhry C, Farr G.W, Todd M.J, Rye H.S, Brunger A.T, Adams P.D, Horwich A.L, Sigler P.B | 2003 | Crystal structure of groEL-groES | 1PCQ; http://dx.doi.org/10.2210/pdb1PCQ/pdb | Publicly available at the RCSB Protein Data Bank. |
| Horsefield R, Yankovskaya V, Sexton G, Whittingham W, Shiomi K, Omura S, Byrne B, Cecchini G, Iwata S | 2005 | Complex II (Succinate Dehydrogenase) From E. Coli with Atpenin A5 inhibitor co-crystallized at the ubiquinone binding site | 2ACZ; http://dx.doi.org/10.2210/pdb2ACZ/pdb | Publicly available at the RCSB Protein Data Bank. |
| Mcevoy M.M, Hausrath A.C, Randolph G.B, Remington S.J, Dahlquist F.W | 1998 | CHEY-BINDING (P2) DOMAIN OF CHEA IN COMPLEX WITH CHEY FROM ESCHERICHIA COLI | 1EAY; http://dx.doi.org/10.2210/pdb1EAY/pdb | Publicly available at the RCSB Protein Data Bank. |
| Bertero M.G, Rothery R.A, Boroumand N, Palak M, Blasco F, Ginet N, Weiner J.H, Strynadka N.C.J | 2004 | The crystal structure of the NarGHI mutant NarI-H66Y | 1Y5L; http://dx.doi.org/10.2210/pdb1Y5L/pdb | Publicly available at the RCSB Protein Data Bank. |
| Borovinskaya M.A, Shoji S, Holton J.M, Fredrick K, Cate J.H.D | 2007 | Crystal structure of the bacterial ribosome from Escherichia coli in complex with spectinomycin. This file contains the 50S subunit of the second 70S ribosome. The entire crystal structure contains two 70S ribosomes | 2QOX; http://dx.doi.org/10.2210/pdb2QOX/pdb | Publicly available at the RCSB Protein Data Bank. |
| Rudolph M.J, Wuebbens M.M, Rajagolpalan K.V, Schindelin H | 2000 | MOLYBDOPTERIN SYNTHASE (MOAD/MOAE) | 1FMA; http://dx.doi.org/10.2210/pdb1FMA/pdb | Publicly available at the RCSB Protein Data Bank. |
| Shibata N | 2009 | Crystal structure of ethanolamine ammonia-lyase from Escherichia coli complexed with CN-Cbl and ethanolamine | 3ABO; http://dx.doi.org/10.2210/pdb3ABO/pdb | Publicly available at the RCSB Protein Data Bank. |
| Stevens R.C, Kantrowitz E.R, Lipscomb W.N | 1992 | ARGININE 54 IN THE ACTIVE SITE OF ESCHERICHIA COLI ASPARTATE TRANSCARBAMOYLASE IS CRITICAL FOR CATALYSIS: A SITE-SPECIFIC MUTAGENESIS, NMR AND X-RAY CRYSTALLOGRAPHY STUDY | 1ACM; http://dx.doi.org/10.2210/pdb1ACM/pdb | Publicly available at the RCSB Protein Data Bank. |

| Heng S, Stieglitz K.A, Eldo J, Xia J, Cardia J.P, Kantrowitz E.R | 2006 | The Structure of Wild-Type E. Coli Aspartate Transcarbamoylase in Complex with Novel T State Inhibitors at 2.10 Resolution | 2FZC; http://dx.doi.org/10.2210/pdb2FZC/pdb | Publicly available at the RCSB Protein Data Bank. |
|---|---|---|---|---|
| Boal A.K, Cotruvo J.A Jnr, Stubbe J, Rosenzweig A.C | 2010 | Ribonucleotide Reductase Dimanganese(II)-NrdF from Escherichia coli in Complex with Reduced NrdI with a Trapped Peroxide | 3N3B; http://dx.doi.org/10.2210/pdb3N3B/pdb | Publicly available at the RCSB Protein Data Bank. |
| Suh M.K, Ku B, Ha N.C, Woo J.S, Oh B.H | 2008 | Crystal Structure of the MukE-MukF Complex | 3EUH; http://dx.doi.org/10.2210/pdb3EUH/pdb | Publicly available at the RCSB Protein Data Bank. |
| Borovinskaya M.A, Shoji S, Holton J.M, Fredrick K, Cate J.H.D | 2007 | Crystal structure of the bacterial ribosome from Escherichia coli in complex with spectinomycin. This file contains the 50S subunit of the first 70S ribosome. The entire crystal structure contains two 70S ribosomes | 2QOV; http://dx.doi.org/10.2210/pdb2QOV/pdb | Publicly available at the RCSB Protein Data Bank. |
| Borovinskaya M.A, Shoji S, Holton J.M, Fredrick K, Cate J.H.D | 2007 | Crystal structure of the bacterial ribosome from Escherichia coli in complex with spectinomycin. This file contains the 30S subunit of the second 70S ribosome, with spectinomycin bound. The entire crystal structure contains two 70S ribosomes | 2QOW; http://dx.doi.org/10.2210/pdb2QOW/pdb | Publicly available at the RCSB Protein Data Bank. |
| Huang J, Lipscomb W.N | 2003 | Products in the T State of Aspartate Transcarbamylase: Crystal Structure of the Phosphate and N-carbamyl-L-aspartate Ligated Enzyme | 1R0C; http://dx.doi.org/10.2210/pdb1R0C/pdb | Publicly available at the RCSB Protein Data Bank. |
| Huang J, Lipscomb W.N | 2003 | Aspartate Transcarbamylase (ATCase) of Escherichia coli: A New Crystalline R State Bound to PALA, or to Product Analogues Phosphate and Citrate | 1R0B; http://dx.doi.org/10.2210/pdb1R0B/pdb | Publicly available at the RCSB Protein Data Bank. |
| Schuwirth B.S, Vila-Sanjurjo A, Cate J.H.D | 2006 | Crystal structure of the bacterial ribosome from escherichia coli in complex with the antibiotic kasugamyin at 3.5A resolution. this file contains the 30s subunit of one 70s ribosome. the entire crystal structure contains two 70s ribosomes and is described in remark 400 | 1VS5; http://dx.doi.org/10.2210/pdb1VS5/pdb | Publicly available at the RCSB Protein Data Bank. |
| Schuwirth B.S, Vila-Sanjurjo A, Cate J.H.D | 2006 | Crystal structure of the bacterial ribosome from escherichia coli in complex with the antibiotic kasugamyin at 3.5A resolution. this file contains the 50s subunit of one 70s ribosome. the entire crystal structure contains two 70s ribosomes and is described in remark 400 | 1VS6; http://dx.doi.org/10.2210/pdb1VS6/pdb | Publicly available at the RCSB Protein Data Bank. |

| | | | | |
|---|---|---|---|---|
| Wada K | 2008 | Crystal structure of SufC-SufD complex involved in the iron-sulfur cluster biosynthesis | 2ZU0; http://dx.doi.org/10.2210/pdb2ZU0/pdb | Publicly available at the RCSB Protein Data Bank. |
| Schuwirth B.S, Vila-Sanjurjo A, Cate J.H.D | 2006 | Crystal structure of the bacterial ribosome from escherichia coli in complex with the antibiotic kasugamyin at 3.5a resolution. this file contains the 50s subunit of one 70s ribosome. The entire crystal structure contains two 70s ribosomes and is described in remark 400 | 1VS8; http://dx.doi.org/10.2210/pdb1VS8/pdb | Publicly available at the RCSB Protein Data Bank. |
| Clore G.M, Garrett D.S, Gronenborn A.M | 1998 | COMPLEX OF THE AMINO TERMINAL DOMAIN OF ENZYME I AND THE HISTIDINE-CONTAINING PHOSPHOCARRIER PROTEIN HPR FROM ESCHERICHIA COLI NMR, RESTRAINED REGULARIZED MEAN STRUCTURE | 3EZE; http://dx.doi.org/10.2210/pdb3EZE/pdb | Publicly available at the RCSB Protein Data Bank. |
| Clore G.M, Garrett D.S, Gronenborn A.M | 1998 | COMPLEX OF THE AMINO TERMINAL DOMAIN OF ENZYME I AND THE HISTIDINE-CONTAINING PHOSPHOCARRIER PROTEIN HPR FROM ESCHERICHIA COLI | 3EZB; http://dx.doi.org/10.2210/pdb3EZB/pdb | Publicly available at the RCSB Protein Data Bank. |
| Clore G.M, Garrett D.S, Gronenborn A.M | 1998 | COMPLEX OF THE AMINO TERMINAL DOMAIN OF ENZYME I AND THE HISTIDINE-CONTAINING PHOSPHOCARRIER PROTEIN HPR FROM ESCHERICHIA COLI NMR, RESTRAINED REGULARIZED MEAN STRUCTURE | 3EZA; http://dx.doi.org/10.2210/pdb3EZA/pdb | Publicly available at the RCSB Protein Data Bank. |
| Rastogi V.K, Girvin M.E | 1999 | A1C12 SUBCOMPLEX OF F1FO ATP SYNTHASE | 1C17; http://dx.doi.org/10.2210/pdb1C17/pdb | Publicly available at the RCSB Protein Data Bank. |
| Su C.-C | 2010 | Crystal structure of the CusBA heavy-metal efflux complex from Escherichia E. coli | 3NE5; http://dx.doi.org/10.2210/pdb3NE5/pdb | Publicly available at the RCSB Protein Data Bank. |
| Zhang W, Dunkle J.A, Cate J.H.D | 2009 | Crystal structure of the E. coli 70S ribosome in an intermediate state of ratcheting | 3I1T; http://dx.doi.org/10.2210/pdb3I1T/pdb | Publicly available at the RCSB Protein Data Bank. |
| Dunkle J.A, Zhang W, Cate J.H.D, Mankin A.S | 2010 | Crystal structure of the E. coli ribosome bound to CEM-101. This file contains the 30S subunit of the second 70S ribosome | 3ORA; http://dx.doi.org/10.2210/pdb3ORA/pdb | Publicly available at the RCSB Protein Data Bank. |
| Grishkovskaya I, Bonsor D.A, Kleanthous C, Dodson E.J | 2006 | Crystal structure of TolB/Pal complex | 2HQS; http://dx.doi.org/10.2210/pdb2HQS/pdb | Publicly available at the RCSB Protein Data Bank. |

| | | | | |
|---|---|---|---|---|
| Shibata N | 2009 | Crystal structure of ethanolamine ammonia-lyase from Escherichia coli complexed with CN-Cbl and 2-amino-1-propanol | 3ABQ; http://dx.doi.org/10.2210/pdb3ABQ/pdb | Publicly available at the RCSB Protein Data Bank. |
| Graille M, Heurgue-Hamard V, Champ S, Mora L, Scrima N, Ulryck N, van Tilbeurgh H, Buckingham R.H | 2005 | Molecular basis for bacterial class 1 release factor methylation by PrmC | 2B3T; http://dx.doi.org/10.2210/pdb2B3T/pdb | Publicly available at the RCSB Protein Data Bank. |
| Kim D.Y, Kwon E, Choi J.K, Hwang H.-Y, Kim K.K | 2010 | Structural basis for the negative regulation of bacterial stress response by RseB | 3M4W; http://dx.doi.org/10.2210/pdb3M4W/pdb | Publicly available at the RCSB Protein Data Bank. |
| Zhang W, Dunkle J.A, Cate J.H.D | 2009 | Crystal structure of the E. coli 70S ribosome in an intermediate state of ratcheting | 3I1R; http://dx.doi.org/10.2210/pdb3I1R/pdb | Publicly available at the RCSB Protein Data Bank. |
| Borovinskaya M.A, Pai R.D, Zhang W, Schuwirth B.-S, Holton J.M, Hirokawa G, Kaji H, Kaji A, Cate J.H.D | 2007 | Crystal structure of the bacterial ribosome from Escherichia coli in complex with neomycin. This file contains the 50S subunit of the second 70S ribosome, with neomycin bound. The entire crystal structure contains two 70S ribosomes and is described in remark 400 | 2QAO; http://dx.doi.org/10.2210/pdb2QAO/pdb | Publicly available at the RCSB Protein Data Bank. |
| Thoden J.B, Huang X, Raushel F.M, Holden H.M | 1999 | CRYSTAL STRUCTURE OF THE CARBAMOYL PHOSPHATE SYNTHETASE: SMALL SUBUNIT MUTANT C269S WITH BOUND GLUTAMINE | 1C3O; http://dx.doi.org/10.2210/pdb1C3O/pdb | Publicly available at the RCSB Protein Data Bank. |
| Tomasiak T.M, Maklashina E, Cecchini G, Iverson T.M | 2008 | E. coli Quinol fumarate reductase FrdA T234A mutation | 3CIR; http://dx.doi.org/10.2210/pdb3CIR/pdb | Publicly available at the RCSB Protein Data Bank. |
| Oldham M.L, Chen J | 2011 | Crystal structure of the maltose-binding protein/maltose transporter complex in an outward-facing conformation bound to MgAMPPNP | 3RLF; http://dx.doi.org/10.2210/pdb3RLF/pdb | Publicly available at the RCSB Protein Data Bank. |
| Ruprecht J, Yankovskaya V, Maklashina E, Iwata S, Cecchini G | 2009 | Crystal structure of the E. coli succinate:quinone oxidoreductase (SQR) SdhC His84Met mutant | 2WU2; http://dx.doi.org/10.2210/pdb2WU2/pdb | Publicly available at the RCSB Protein Data Bank. |
| Ke H, Lipscomb W.N, Cho Y, Honzatko R.B | 1989 | COMPLEX OF N-PHOSPHONACETYL-L-ASPARTATE WITH ASPARTATE CARBAMOYL-TRANSFERASE. X-RAY REFINEMENT, ANALYSIS OF CONFORMATIONAL CHANGES AND CATALYTIC AND ALLOSTERIC MECHANISMS | 8ATC; http://dx.doi.org/10.2210/pdb8ATC/pdb | Publicly available at the RCSB Protein Data Bank. |
| Bertero M.G, Rothery R.A, Boroumand N, Palak M, Blasco F, Ginet N, Weiner J.H, Strynadka N.C.J | 2004 | The crystal structure of the NarGHI mutant NarI-K86A in complex with pentachlorophenol | 1Y5N; http://dx.doi.org/10.2210/pdb1Y5N/pdb | Publicly available at the RCSB Protein Data Bank. |

| | | | | |
|---|---|---|---|---|
| Woo J.S, Lim J.H, Shin H.C, Oh B.H | 2008 | Crystal structure of MukE-MukF(residues 292-443)-MukB(head domain)-ATPgammaS complex, asymmetric dimer | 3EUK; http://dx.doi.org/10.2210/pdb3EUK/pdb | Publicly available at the RCSB Protein Data Bank. |
| Rees D.C, Kaiser J.T, Kadaba N.S, Johnson E, Lee A.T | 2008 | Crystal structure of methionine importer MetNI | 3DHW; http://dx.doi.org/10.2210/pdb3DHW/pdb | Publicly available at the RCSB Protein Data Bank. |
| Stieglitz K, Stec B, Baker D.P, Kantrowitz E.R | 2004 | Aspartate Transcarbamoylase Catalytic Chain Mutant Glu50Ala Complexed with N-(Phosphonacetyl-L-Aspartate) (PALA) | 1TTH; http://dx.doi.org/10.2210/pdb1TTH/pdb | Publicly available at the RCSB Protein Data Bank. |
| Stieglitz K.A, Alam N, Xia J, Gourinath S, Tsuruta H, Kantrowitz E.R | 2004 | The Structure of E. coli Aspartate Transcarbamoylase Q137A Mutant in The R-State | 1XJW; http://dx.doi.org/10.2210/pdb1XJW/pdb | Publicly available at the RCSB Protein Data Bank. |
| Guo F, Esser L, Singh S.K, Maurizi M.R, Xia D | 2002 | CRYSTAL STRUCTURE ANALYSIS OF ClpSN WITH TRANSITION METAL ION BOUND | 1MBX; http://dx.doi.org/10.2210/pdb1MBX/pdb | Publicly available at the RCSB Protein Data Bank. |
| Mendes K.R, Kantrowitz E.R | 2010 | Crystal structure of the C47A/A241C disulfide-linked E. coli Aspartate Transcarbamoylase holoenzyme | 3MPU; http://dx.doi.org/10.2210/pdb3MPU/pdb | Publicly available at the RCSB Protein Data Bank. |
| Jin L, Stec B, Kantrowitz E.R | 2000 | CRYSTAL STRUCTURE OF E. COLI ASPARTATE TRANSCARBAMOYLASE P268A MUTANT IN THE T-STATE | 1EZZ; http://dx.doi.org/10.2210/pdb1EZZ/pdb | Publicly available at the RCSB Protein Data Bank. |
| Dunkle J.A, Xiong L, Mankin A.S, Cate J.H.D | 2010 | Crystal structure of the E. coli ribosome bound to telithromycin. This file contains the 50S subunit of the first 70S ribosome with telithromycin bound | 3OAT; http://dx.doi.org/10.2210/pdb3OAT/pdb | Publicly available at the RCSB Protein Data Bank. |
| Grigoriu S, Brown B.L, Arruda J.M, Peti W, Page R | 2009 | Structure of the N-terminal domain of the E. coli antitoxin MqsA (YgiT/b3021) in complex with the E. coli toxin MqsR (YgiU/b3022) | 3HI2; http://dx.doi.org/10.2210/pdb3HI2/pdb | Publicly available at the RCSB Protein Data Bank. |
| Guo F, Esser L, Singh S.K, Maurizi M.R, Xia D | 2002 | Crystal Structure Analysis of ClpSN heterodimer | 1MBU; http://dx.doi.org/10.2210/pdb1MBU/pdb | Publicly available at the RCSB Protein Data Bank. |
| Guo F, Esser L, Singh S.K, Maurizi M.R, Xia D | 2002 | CRYSTAL STRUCTURE ANALYSIS OF ClpSN HETERODIMER TETRAGONAL FORM | 1MBV; http://dx.doi.org/10.2210/pdb1MBV/pdb | Publicly available at the RCSB Protein Data Bank. |
| Dunkle J.A, Xiong L, Mankin A.S, Cate J.H.D | 2010 | Crystal structure of the E. coli ribosome bound to telithromycin. This file contains the 30S subunit of the first 70S ribosome | 3OAQ; http://dx.doi.org/10.2210/pdb3OAQ/pdb | Publicly available at the RCSB Protein Data Bank. |
| Okamura-Ikeda K, Hosaka H | 2009 | Crystal Structure of ET-EHred-5-CH3-THF complex | 3A8I; http://dx.doi.org/10.2210/pdb3A8I/pdb | Publicly available at the RCSB Protein Data Bank. |
| Okamura-Ikeda K, Hosaka H | 2009 | Crystal Structure of ET-EHred complex | 3A8J; http://dx.doi.org/10.2210/pdb3A8J/pdb | Publicly available at the RCSB Protein Data Bank. |
| Okamura-Ikeda K, Hosaka H | 2009 | Crystal Structure of ETD97N-EHred complex | 3A8K; http://dx.doi.org/10.2210/pdb3A8K/pdb | Publicly available at the RCSB Protein Data Bank. |

| | | | | |
|---|---|---|---|---|
| Campbell E.A, Tupy J.L, Gruber T.M, Wang S, Sharp M.M, Gross C.A, Darst S.A | 2003 | Crystal Structure of Escherichia coli sigmaE with the Cytoplasmic Domain of its Anti-sigma RseA | 1OR7; http://dx.doi.org/10.2210/pdb1OR7/pdb | Publicly available at the RCSB Protein Data Bank. |
| Ruprecht J, Yankovskaya V, Maklashina E, Iwata S, Cecchini G | 2009 | Crystal structure of the E. coli succinate:quinone oxidoreductase ( SQR) SdhD Tyr83Phe mutant | 2WS3; http://dx.doi.org/10.2210/pdb2WS3/pdb | Publicly available at the RCSB Protein Data Bank. |
| Thoden J.B, Raushel F.M, Holden H.M | 1999 | CARBAMOYL PHOSPHATE SYNTHETASE FROM ESCHERICHIS COLI WITH COMPLEXED WITH THE ALLOSTERIC LIGAND IMP | 1CE8; http://dx.doi.org/10.2210/pdb1CE8/pdb | Publicly available at the RCSB Protein Data Bank. |
| Thoden J.B, Huang X, Raushel F.M, Holden H.M | 2002 | Crystal Structure of the G359F (small subunit) Point Mutant of Carbamoyl Phosphate Synthetase | 1M6V; http://dx.doi.org/10.2210/pdb1M6V/pdb | Publicly available at the RCSB Protein Data Bank. |
| Hvorup R.N, Goetz B.A, Niederer M, Hollenstein K, Perozo E, Locher K.P | 2007 | ABC-transporter BtuCD in complex with its periplasmic binding protein BtuF | 2QI9; http://dx.doi.org/10.2210/pdb2QI9/pdb | Publicly available at the RCSB Protein Data Bank. |
| Bertero M.G, Rothery R.A, Boroumand N, Palak M, Blasco F, Ginet N, Weiner J.H, Strynadka N.C.J | 2004 | The crystal structure of the NarGHI mutant NarI-K86A | 1Y5I; http://dx.doi.org/10.2210/pdb1Y5I/pdb | Publicly available at the RCSB Protein Data Bank. |
| Dunkle J.A, Xiong L, Mankin A.S, Cate J.H.D | 2010 | Crystal structure of the E. coli ribosome bound to erythromycin. This file contains the 30S subunit of the first 70S ribosome | 3OFO; http://dx.doi.org/10.2210/pdb3OFO/pdb | Publicly available at the RCSB Protein Data Bank. |
| Locher K.P, Lee A.T, Rees D.C | 2002 | Bacterial ABC Transporter Involved in B12 Uptake | 1L7V; http://dx.doi.org/10.2210/pdb1L7V/pdb | Publicly available at the RCSB Protein Data Bank. |
| Wilce M.C.J, Rodgers A.J.W | 2000 | COMPLEX OF GAMMA/EPSILON ATP SYNTHASE FROM E.COLI | 1FS0; http://dx.doi.org/10.2210/pdb1FS0/pdb | Publicly available at the RCSB Protein Data Bank. |
| Casino P, Marina A | 2008 | Structure of a histidine kinase-response regulator complex reveals insights into Two-component signaling and a novel cis-autophosphorylation mechanism | 3DGE; http://dx.doi.org/10.2210/pdb3DGE/pdb | Publicly available at the RCSB Protein Data Bank. |
| Ruprecht J, Yankovskaya V, Maklashina E, Iwata S, Cecchini G | 2009 | Crystal structure of the E. coli succinate:quinone oxidoreductase (SQR) SdhD His71Met mutant | 2WU5; http://dx.doi.org/10.2210/pdb2WU5/pdb | Publicly available at the RCSB Protein Data Bank. |
| Stieglitz K, Stec B, Baker D.P, Kantrowitz E.R | 2004 | Aspartate Transcarbamoylase Catalytic Chain Mutant E50A Complex with Phosphonoacetamide | 1TU0; http://dx.doi.org/10.2210/pdb1TU0/pdb | Publicly available at the RCSB Protein Data Bank. |
| Dunkle J.A, Xiong L, Mankin A.S, Cate J.H.D | 2010 | Crystal structure of the E. coli ribosome bound to telithromycin. This file contains the 50S subunit of the second 70S ribosome | 3OAS; http://dx.doi.org/10.2210/pdb3OAS/pdb | Publicly available at the RCSB Protein Data Bank. |
| Shi R, McDonald L, Matte A, Cygler M, Ekiel I, Montreal-Kingston Bacterial Structural Genomics Initiative (BSGI) | 2010 | Crystal Structure of E.coli Dha kinase DhaK-DhaL complex | 3PNL; http://dx.doi.org/10.2210/pdb3PNL/pdb | Publicly available at the RCSB Protein Data Bank. |

| | | | |
|---|---|---|---|
| Borovinskaya M.A, Shoji S, Holton J.M, Fredrick K, Cate J.H.D | 2007 | Crystal structure of the bacterial ribosome from Escherichia coli in complex with spectinomycin and neomycin. This file contains the 50S subunit of the second 70S ribosome, with neomycin bound. The entire crystal structure contains two 70S ribosomes | 2QP1; http://dx.doi.org/ 10.2210/pdb2QP1/pdb | Publicly available at the RCSB Protein Data Bank. |
| Dunkle J.A, Xiong L, Mankin A.S, Cate J.H.D | 2010 | Crystal structure of the E. coli ribosome bound to telithromycin. This file contains the 30S subunit of the second 70S ribosome | 3OAR; http://dx.doi.org/ 10.2210/pdb3OAR/pdb | Publicly available at the RCSB Protein Data Bank. |
| Borovinskaya M.A, Pai R.D, Zhang W, Schuwirth B.-S, Holton J.M, Hirokawa G, Kaji H, Kaji A, Cate J.H.D | 2007 | Crystal structure of the bacterial ribosome from Escherichia coli in complex with ribosome recycling factor (RRF). This file contains the 50S subunit of the first 70S ribosome, with RRF bound. The entire crystal structure contains two 70S ribosomes and is described in remark 400 | 2QBE; http://dx.doi.org/ 10.2210/pdb2QBE/pdb | Publicly available at the RCSB Protein Data Bank. |
| Kosman R.P, Gouaux J.E, Lipscomb W.N | 1992 | CRYSTAL STRUCTURE OF CTP-LIGATED T STATE ASPARTATE TRANSCARBAMOYLASE AT 2.5 ANGSTROMS RESOLUTION: IMPLICATIONS FOR ATCASE MUTANTS AND THE MECHANISM OF NEGATIVE COOPERATIVITY | 1RAB; http://dx.doi.org/ 10.2210/pdb1RAB/pdb | Publicly available at the RCSB Protein Data Bank. |
| Kosman R.P, Gouaux J.E, Lipscomb W.N | 1992 | CRYSTAL STRUCTURE OF CTP-LIGATED T STATE ASPARTATE TRANSCARBAMOYLASE AT 2.5 ANGSTROMS RESOLUTION: IMPLICATIONS FOR ATCASE MUTANTS AND THE MECHANISM OF NEGATIVE COOPERATIVITY | 1RAC; http://dx.doi.org/ 10.2210/pdb1RAC/pdb | Publicly available at the RCSB Protein Data Bank. |
| Kosman R.P, Gouaux J.E, Lipscomb W.N | 1992 | CRYSTAL STRUCTURE OF CTP-LIGATED T STATE ASPARTATE TRANSCARBAMOYLASE AT 2.5 ANGSTROMS RESOLUTION: IMPLICATIONS FOR ATCASE MUTANTS AND THE MECHANISM OF NEGATIVE COOPERATIVITY | 1RAA; http://dx.doi.org/ 10.2210/pdb1RAA/pdb | Publicly available at the RCSB Protein Data Bank. |
| Kosman R.P, Gouaux J.E, Lipscomb W.N | 1992 | CRYSTAL STRUCTURE OF CTP-LIGATED T STATE ASPARTATE TRANSCARBAMOYLASE AT 2.5 ANGSTROMS RESOLUTION: IMPLICATIONS FOR ATCASE MUTANTS AND THE MECHANISM OF NEGATIVE COOPERATIVITY | 1RAF; http://dx.doi.org/ 10.2210/pdb1RAF/pdb | Publicly available at the RCSB Protein Data Bank. |

| | | | | |
|---|---|---|---|---|
| Kosman R.P, Gouaux J.E, Lipscomb W.N | 1992 | CRYSTAL STRUCTURE OF CTP-LIGATED T STATE ASPARTATE TRANSCARBAMOYLASE AT 2.5 ANGSTROMS RESOLUTION: IMPLICATIONS FOR ATCASE MUTANTS AND THE MECHANISM OF NEGATIVE COOPERATIVITY | 1RAG; http://dx.doi.org/10.2210/pdb1RAG/pdb | Publicly available at the RCSB Protein Data Bank. |
| Kosman R.P, Gouaux J.E, Lipscomb W.N | 1992 | CRYSTAL STRUCTURE OF CTP-LIGATED T STATE ASPARTATE TRANSCARBAMOYLASE AT 2.5 ANGSTROMS RESOLUTION: IMPLICATIONS FOR ATCASE MUTANTS AND THE MECHANISM OF NEGATIVE COOPERATIVITY | 1RAD; http://dx.doi.org/10.2210/pdb1RAD/pdb | Publicly available at the RCSB Protein Data Bank. |
| Kosman R.P, Gouaux J.E, Lipscomb W.N | 1992 | CRYSTAL STRUCTURE OF CTP-LIGATED T STATE ASPARTATE TRANSCARBAMOYLASE AT 2.5 ANGSTROMS RESOLUTION: IMPLICATIONS FOR ATCASE MUTANTS AND THE MECHANISM OF NEGATIVE COOPERATIVITY | 1RAE; http://dx.doi.org/10.2210/pdb1RAE/pdb | Publicly available at the RCSB Protein Data Bank. |
| Jormakka M, Tornroth S, Byrne B, Iwata S | 2002 | FORMATE DEHYDROGENASE N FROM E. COLI | 1KQG; http://dx.doi.org/10.2210/pdb1KQG/pdb | Publicly available at the RCSB Protein Data Bank. |
| Jormakka M, Tornroth S, Byrne B, Iwata S | 2002 | FORMATE DEHYDROGENASE N FROM E. COLI | 1KQF; http://dx.doi.org/10.2210/pdb1KQF/pdb | Publicly available at the RCSB Protein Data Bank. |
| Kosman R.P, Gouaux J.E, Lipscomb W.N | 1992 | CRYSTAL STRUCTURE OF CTP-LIGATED T STATE ASPARTATE TRANSCARBAMOYLASE AT 2.5 ANGSTROMS RESOLUTION: IMPLICATIONS FOR ATCASE MUTANTS AND THE MECHANISM OF NEGATIVE COOPERATIVITY | 1RAH; http://dx.doi.org/10.2210/pdb1RAH/pdb | Publicly available at the RCSB Protein Data Bank. |
| Kosman R.P, Gouaux J.E, Lipscomb W.N | 1992 | CRYSTAL STRUCTURE OF CTP-LIGATED T STATE ASPARTATE TRANSCARBAMOYLASE AT 2.5 ANGSTROMS RESOLUTION: IMPLICATIONS FOR ATCASE MUTANTS AND THE MECHANISM OF NEGATIVE COOPERATIVITY | 1RAI; http://dx.doi.org/10.2210/pdb1RAI/pdb | Publicly available at the RCSB Protein Data Bank. |
| Gouaux J.E, Stevens R.C, Lipscomb W.N | 1989 | CRYSTAL STRUCTURES OF ASPARTATE CARBAMOYLTRANSFERASE LIGATED WITH PHOSPHONOACETAMIDE, MALONATE, AND CTP OR ATP AT 2.8-ANGSTROMS RESOLUTION AND NEUTRAL P*H | 8AT1; http://dx.doi.org/10.2210/pdb8AT1/pdb | Publicly available at the RCSB Protein Data Bank. |

| | | | | |
|---|---|---|---|---|
| Chaudhry C, Horwich A.L, Brunger A.T, Adams P.D | 2004 | Crystal structure of GroEL14-GroES7-(ADP-AlFx)7 | 1SVT; http://dx.doi.org/10.2210/pdb1SVT/pdb | Publicly available at the RCSB Protein Data Bank. |
| Oldham M.L, Chen J | 2010 | Crystal Structure of a pre-translocation state MBP-Maltose transporter complex without nucleotide | 3PV0; http://dx.doi.org/10.2210/pdb3PV0/pdb | Publicly available at the RCSB Protein Data Bank. |
| Xia D, Maurizi M.R, Guo F, Singh S.K, Esser L | 2003 | ATP-dependent Clp protease ATP-binding subunit clpA/ATP-dependent Clp protease adaptor protein clpS | 1R6O; http://dx.doi.org/10.2210/pdb1R6O/pdb | Publicly available at the RCSB Protein Data Bank. |
| Zhou J, Lancaster L, Trakhanov S, Noller H.F | 2011 | Crystal Structure of Release Factor RF3 Trapped in the GTP State on a Rotated Conformation of the Ribosome (without viomycin) | 3UOS; http://dx.doi.org/10.2210/pdb3UOS/pdb | Publicly available at the RCSB Protein Data Bank. |
| Singleton M.R, Dillingham M.S, Gaudier M, Kowalczykowski S.C, Wigley D.B | 2004 | RecBCD: DNA complex | 1W36; http://dx.doi.org/10.2210/pdb1W36/pdb | Publicly available at the RCSB Protein Data Bank. |
| Borovinskaya M.A, Pai R.D, Zhang W, Schuwirth BS, Holton J.M, Hirokawa G, Kaji H, Kaji A, Cate J.H.D | 2007 | Crystal structure of the bacterial ribosome from Escherichia coli in complex with ribosome recycling factor (RRF). This file contains the 30S subunit of the first 70S ribosome. The entire crystal structure contains two 70S ribosomes and is described in remark 400 | 2QBD; http://dx.doi.org/10.2210/pdb2QBD/pdb | Publicly available at the RCSB Protein Data Bank. |
| Gouaux J.E, Lipscomb W.N | 1989 | CRYSTAL STRUCTURES OF PHOSPHONOACETAMIDE LIGATED T AND PHOSPHONOACETAMIDE AND MALONATE LIGATED R STATES OF ASPARTATE CARBAMOYLTRANSFERASE AT 2.8-ANGSTROMS RESOLUTION AND NEUTRAL PH | 3AT1; http://dx.doi.org/10.2210/pdb3AT1/pdb | Publicly available at the RCSB Protein Data Bank. |
| Shomura Y, Higuchi Y | 2012 | Crystal structure of HypE-HypF complex | 3VTI; http://dx.doi.org/10.2210/pdb3VTI/pdb | Publicly available at the RCSB Protein Data Bank. |
| Zeth K, Ravelli R.B, Paal K, Cusack S, Bukau B, Dougan D.A | 2002 | Structural basis of ClpS-mediated switch in ClpA substrate recognition | 1LZW; http://dx.doi.org/10.2210/pdb1LZW/pdb | Publicly available at the RCSB Protein Data Bank. |
| Berk V, Zhang W, Pai R.D, Cate J.H.D | 2006 | Crystal Structure of Ribosome with messenger RNA and the Anticodon stem-loop of P-site tRNA. This file contains the 50s subunit of one 70s ribosome. The entire crystal structure contains two 70s ribosomes and is described in remark 400 | 2I2T; http://dx.doi.org/10.2210/pdb2I2T/pdb | Publicly available at the RCSB Protein Data Bank. |

| Borovinskaya M.A, Shoji S, Holton J.M, Fredrick K, Cate J.H.D | 2007 | Crystal structure of the bacterial ribosome from Escherichia coli in complex with spectinomycin and neomycin. This file contains the 30S subunit of the first 70S ribosome, with spectinomycin and neomycin bound. The entire crystal structure contains two 70S ribosomes | 2QOY; http://dx.doi.org/ 10.2210/pdb2QOY/pdb | Publicly available at the RCSB Protein Data Bank. |
|---|---|---|---|---|
| Wang J, Stieglitz K.A, Cardia J.P, Kantrowitz E.R | 2005 | Structure of wild-type E. coli Aspartate Transcarbamoylase in the presence of CTP, carbamoyl phosphate at 2.50 A resolution | 1ZA2; http://dx.doi.org/ 10.2210/pdb1ZA2/pdb | Publicly available at the RCSB Protein Data Bank. |
| Borovinskaya M.A, Pai R.D, Zhang W, Schuwirth BS, Holton JM, Hirokawa G, Kaji H, Kaji A, Cate J.H.D | 2007 | Crystal structure of the bacterial ribosome from Escherichia coli in complex with ribosome recycling factor (RRF). This file contains the 50S subunit of the second 70S ribosome, with RRF bound. The entire crystal structure contains two 70S ribosomes and is described in remark 400 | 2QBG; http://dx.doi.org/ 10.2210/pdb2QBG/pdb | Publicly available at the RCSB Protein Data Bank. |
| Wang J, Stieglitz K.A, Cardia J.P, Kantrowitz E.R | 2005 | Structure of wild-type E. coli Aspartate Transcarbamoylase in the presence of CTP at 2.20 A resolution | 1ZA1; http://dx.doi.org/ 10.2210/pdb1ZA1/pdb | Publicly available at the RCSB Protein Data Bank. |
| Thoden J.B, Huang X, Raushel FM, Holden H.M | 1999 | CRYSTAL STRUCTURE OF CARBAMOYL PHOSPHATE SYNTHETASE: SMALL SUBUNIT MUTATION C269S | 1C30; http://dx.doi.org/ 10.2210/pdb1C30/pdb | Publicly available at the RCSB Protein Data Bank. |
| Williams M.K, Stec B, Kantrowitz E.R | 1998 | ASPARTATE TRANSCARBOMYLASE REGULATORY CHAIN MUTANT (T82A) | 1NBE; http://dx.doi.org/ 10.2210/pdb1NBE/pdb | Publicly available at the RCSB Protein Data Bank. |

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
