## [Decision Letter]

Thank you for sending your work entitled “Sequence co-evolution gives 3D contacts and structures of protein complexes”““ for consideration at *eLife.* Your article has been favorably evaluated by John Kuriyan (Senior editor), working with a member of our Board of Reviewing Editors, and 3 reviewers.

The editors and the other reviewers discussed their comments before we reached this decision, and the Reviewing editor has assembled the following comments to help you prepare a revised submission.

This manuscript describes an approach to predict 3D residue-residue contacts at protein interfaces from an analysis of multiple sequence alignments. The described computational approach (using sequence co-evolution to predict contacts between interacting proteins and to generate three-dimensional models of complex structures) appears to be quite successful and is an important contribution that is of interest to a broad audience. The analyses, results, and conclusions are similar to work conducted during the same period by Baker & coworkers, published recently in *eLife.* Nevertheless, it is the consensus opinion of the reviewers that the present manuscript is potentially suitable for publication in *eLife* as well, provided that all of the comments raised by the reviewers can be addressed satisfactorily.

We have appended the comments from the three reviewers below. Although many individual points are raised, you will see that the principal concern of the reviewers is that the manuscript falls short of providing the reader with sufficient analysis to judge the validity of the conclusions. It should be possible for you to revise the manuscript to address these concerns without much in the way of new calculations, so we hope that it should be relatively straightforward to deal with these issues.

In addition, since the Baker paper came out recently, it should be given appropriate credit in a revised version. We recommend removing the word “new” from the manuscript for this reason, and instead simply state what was done.

Reviewer 1:

1) The Abstract claims that the authors' approach can discriminate between interacting and non-interacting proteins. However, as far as I can see, the rest of the paper concerns only calculating the interface residues of proteins which are known to interact. No evidence is presented to support the claim to be able to calculate non-interactors. The authors' claim should be corrected accordingly.

2) The authors claim that the co-evolutionary approach has not previously been applied to prediction of protein-protein interfaces. This is incorrect, as the authors seem to be unaware of the previous work of Raphael Guerois' group, which seems rather surprising. The authors should acknowledge the prior work of Faure et al (2012) and Andreani et al. (2013), and they should modify their claim to novelty accordingly. It would also be appropriate to cite properly the prior work of the Baker group (Ovchinnikov et al., 2014) in the same paragraph (the authors currently mention this work only as a “note added during submission”), and the recent review by Andreani and Guerois (2014).

3) In several places the authors say that their approach exploits the fact that interacting proteins are often coded close to each other in a genome. Why is this assumption necessary? According to the description given in Methods, any pair of sequences which are presumed to interact could be concatenated and then used in the authors method. Please clarify.

4) If I understand the authors’ method correctly, they first concatenate the multiple sequences of the presumed interactors, and they then use their previous approach for detecting interacting pairs of columns in the multiple alignment. In this case, when concatenating the alignments for protein “A” with those of protein “B”, the “intra-EC” pairs would appear as A-A and B-B pairs, while the “inter-EC” pairs would come from A-B or B-A interactions. My question is then, are there any observable differences in the inter and intra scores? One might expect that the conservation scores would be lower for inter interactions than intra interactions, as an interface might show more “plasticity” than the core of a domain. It would be very interesting if the authors could comment on this, preferably supported by numerical results.

5) It is incorrect for the authors to refer to their own data set as a “benchmark”. Suggestion: remove all references to the term “benchmark”. However, if the authors make all their data available in a convenient way on-line, it could be proposed as a new “benchmark”.

6) It is wrong to claim that “Experimental evidence... agrees with EVcomplex predictions” (!). Please rewrite this to say that your predictions agree with the experimental evidence. But, if this is your claim, please also support it in some way. What is the experimental evidence? How to you calculate and quantify “agrees with”?

7) Same point in the main text. “The resulting model is consistent with the topologies from cross-linking”. How do you quantify consistent? RMSD from the earlier models? Please provide supporting details.

Reviewer 2:

My main comments concern the way the data in the current manuscript are analyzed and presented. While the main Figures / Table 1 currently illustrate useful examples, they should also show overall analyses of the results over all cases in the benchmark datasets. In several cases this is essential to support the stated conclusions. It does not seem that this would require a substantial amount of new work or lengthy simulations, as most of the relevant data are in the Supplementary Materials (most data are in raw table format or with a separate Figure for each example, which makes it hard for the reader to gauge overall performance – this could be presented in a more easily digestible format in the main text). Specifically, I feel the paper could be substantially improved if the following points were addressed:

1) Please support the statement “Benchmark calculations here ... indicate that the number of sequences in the alignment is critical...” with a Figure showing overall performance versus # of sequences (Figure 3 shows dependency of accuracy on relative rank for only two examples).

2) Show a main Figure comparing predicted contacts against residue distances in the 30 known crystal structures of complexes that had a sufficient number of sequences (Figure 2 only shows contact plots for two examples, and structural pictures for several more, but no overall quantification). This type of analysis could also help to support the statement “For the top ranked benchmark complexes, the majority of the top 5 ECs is correct to within 8A...” with a more quantitative analysis (Table 1 only shows 7 of the 30 complexes in the dataset; [Supplementary-material SD3-data] shows all individual contact maps, but no overall quantification).

3) “... these benchmarks show ... and demonstrate the criteria needed for successful prediction of unknown interactions”. To support this statement, it would be useful to have a main Figure / Table for each important criterion, analyzed over an entire benchmark set (as for the number of sequences in point 1).

4) Please add metrics to support statements that ECs are “completely consistent” with crosslinking data, or that a coupling “coincides with experimental evidence”. The comparison Table in Supplementary Figure 6 could be presented in more quantitative terms in the main text (for example, what does “crosslinking neighborhood” mean?).

5) What are the possible reasons for false-positive co-variation that does not correspond to contact?

6) The manuscript deals entirely with pairwise co-evolution, and seems to neglect the possibility of higher-order complexity in the sequence pattern.

Reviewer 3:

The manuscript presents an upbeat and optimistic view of the success of the method, with a relatively small amount of critical discussion and little if any investigation of what one can learn from cases in which it does not perform as expected. (For example, a list of possible reasons for false-positive co-variation that does not correspond to contact is given, but an analysis of actual instances is not.) Likewise, a set of methods is presented that is relatively particular, and the reason why these specific methods were chosen over other possibilities is not provided. Moreover, the manuscript deals entirely with pairwise co-evolution, and seems to neglect the possibility of higher-order complexity in the sequence pattern. The manuscript could be substantially strengthened by adding analyses and insights in these issues.

---

## [Author Response]

Thank you for your comments and forwarded reviews of our paper. The revised paper addresses the questions from the reviewers. As you point out, the principal request of the reviewers was to see more quantitative analysis to judge the validity of the conclusions. This is addressed by detailed additions to the paper including:

1) A quantitative analysis of the accuracy with summary statistics (Figure 2, with Supplementary figures, tables and data for each individual complex).

2) The use of the EVcomplex scores for inter-protein ECs that allow comparison across proteins (Figure 2, Materials and methods, Supplementary files).

3) A fuller analysis and clearer explanation of how the scores are used to distinguish interactions from non-interactions with a proof of principle exercise of all ATP synthase subunits against all others (a total of 28 interactions), resulting in 86% accuracy.

4) Detailed analysis of agreement of cross-linking experiments and predictions for a –subunit ECs and a- to b- subunit inter-ECs, [Supplementary-material SD6-data] and Figure 7.Author response image 1.

The results are essentially the same as in the original submission and we have now provided the reader with comprehensive access to the numerical results, including sequence alignments, list of all contact predictions, precision of predicted contacts with respect to known 3D structures for all complexes, docked models for 15 complexes,

Reviewer 1:

*1) The Abstract claims that the authors' approach can discriminate between interacting and non-interacting proteins. However, as far as I can see, the rest of the paper concerns only calculating the interface residues of proteins which are known to interact. No evidence is presented to support the claim to be able to calculate non-interactors. The authors' claim should be corrected accordingly*.

We have updated the manuscript to clarify the way in which the method addresses discrimination between interacting and non-interacting proteins, and give the specific example of all against all in the ATP synthase complex. We suggest that the correct identification of interacting and non-interacting protein subunits of the ATP synthase complex could serve as a 'proof of principle' that scores derived from evolutionary couplings can identify interactions (in addition to the identification of the interacting residues between protein pairs). Since identification of the interactions of the 8 subunits of ATP synthase is the only example we report, we have been careful to talk about this as a proof of principle that may be generalized with further work (changes in Abstract, Figure 5, Materials and Methods, [Supplementary-material SD5-data], Supplementary data). [The EV complex score identifies 24/28 of the interactions as correctly interacting or not.]

*2) The authors claim that the co-evolutionary approach has not previously been applied to prediction of protein-protein interfaces. This is incorrect, as the authors seem to be unaware of the previous work of Raphael Guerois' group, which seems rather surprising. The authors should acknowledge the prior work of Faure et al (2012) and Andreani et al. (2013), and they should modify their claim to novelty accordingly*.

*It would also be appropriate to cite properly the prior work of the Baker group (Ovchinnikov et al., 2014) in the same paragraph (the authors currently mention this work only as a “note added during submission”), and the recent review by Andreani and Guerois (2014)*.

We have now referred to the body of work by the Guerois group in the Introduction. We apologize for the oversight of this very interesting work. We also added more complete referencing to the work of the Valencia group; see response to reviewer 3. The Guerois group has used evolutionary information to improve their machine learnt statistical potentials. Our goal is to ask if we can we identify co-evolving residues (from sequence information alone) across proteins in order to ask whether these co-evolving pairs identify close residues, (1) whether they can be used to determine complexes in 3D and (2) whether we can use this to say if two proteins interact. Hence the approaches we present and those of the Guerois group are nicely synergistic and we hope to work with them to bring these methods together in the future.

*3) In several places the authors say that their approach exploits the fact that interacting proteins are often coded close to each other in a genome. Why is this assumption necessary? According to the description given in Methods, any pair of sequences which are presumed to interact could be concatenated and then used in the authors method. Please clarify*.

Thanks for making it clear that we did not explain this sufficiently well. We have now clarified this in Results, Discussion, and in the Materials and methods.

To summarize: The accuracy of the co-evolutionary analysis relies on the concatenation of the protein pair representing a true interaction. This means that the contact prediction across proteins will only be as good as the input alignment pairs. When there is only one protein family member of both interacting partners in each species, one is guaranteed to get the correct alignment, assuming that it is a real complex in all organisms. Hence, subunits of ATP synthase that have only one copy in the genome, would not require this condition, and one could have simply concatenated the one pair per genome. However, for most protein interactions in *E. coli*, there are not yet enough sequences available to allow the use of just one pair per genome, as the complex may not necessarily occur in all bacteria. In these cases, for the time-being, our approach relies on using paralogous interactions to get sufficient sequences for instance histidine kinase; response regulators, transporters and their ATPases. Since we therefore use several paralogous interactions per complex in each genome, the algorithm requires that the partner needs to be correctly paired, something that we do not necessarily know *a priori*, and for which we use genome proximity as a proxy (as others have done in references 22-25). Clearly this problem will secede for many interactions in the next few years as many thousand more bacterial genomes are sequenced, but the problem is an interesting one that we think will have to be thought about in relation to eukaryotic complexes, too.

*4) If I understand the authors*’ *method correctly, they first concatenate the multiple sequences of the presumed interactors, and they then use their previous approach for detecting interacting pairs of columns in the multiple alignment. In this case, when concatenating the alignments for protein “A” with those of protein “B”, the “intra-EC” pairs would appear as A-A and B-B pairs, while the “inter-EC” pairs would come from A-B or B-A interactions. My question is then, are there any observable differences in the inter and intra scores? One might expect that the conservation scores would be lower for inter interactions than intra interactions, as an interface might show more “plasticity” than the core of a domain. It would be very interesting if the authors could comment on this, preferably supported by numerical results*.

The reviewer makes a good point and one that we have now highlighted more clearly in the text. The inter-EC scores within each complex are almost universally lower than the intra–EC scores as would be expected, Figure 2 and Figure 2—figure supplement 1.

In general we observe that the accuracy of the intra-contacts up to the rank of the first inter contact, is a good indication of how accurate the predicted inter-contacts are likely to be. This means we could have used a measure of the accuracy of the *intra*-protein scores to assess the predicted inter-contacts. We chose to develop a scoring method independent that did not rely on the existence of experimentally determined 3D structures of the monomers. The EV complex score implicitly considers where the inter-EC scores lie in the whole distribution instead, hence excluding those inter-ECs that lie within the noise of the whole distribution, see Materials and Methods ([Disp-formula equ1] and [Disp-formula equ2]), Figure 2, Figure 2—figure supplement 1, Figure 2—figure supplement 2, and Supplementary data 3 “EVcomplex predictions”.

*5) It is incorrect for the authors to refer to their own data set as a "benchmark". Suggestion: remove all references to the term "benchmark". However, if the authors make all their data available in a convenient way on-line, it could be proposed as a new "benchmark"*.

We have removed the references to the set of known 3D interactions as ‘benchmark’ and proposed this as a set the community could use in the future, as the reviewer suggests (Materials and methods).

6) It is wrong to claim that "Experimental evidence... agrees with EVcomplex predictions" (!). Please rewrite this to say that your predictions agree with the experimental evidence. But, if this is your claim, please also support it in some way. What is the experimental evidence? How to you calculate and quantify "agrees with"?

We have removed this sentence, and reworded this to reflect the fact we are using the experimental data to support the EC predictions and not the other way around.

*7) Same point in the main text. "The resulting model is consistent with the topologies from cross-linking". How do you quantify consistent? RMSD from the earlier models? Please provide supporting details*.

To our knowledge there are no experimentally determined atomic resolution structures of the a-subunit of ATP synthase. A 3D model of the a-subunit is from 1999 (PDB ID: 1c17) and was computed using five helical–helical interactions that were inferred from second suppressor mutation experiments, and then imposed as distance restraints for TMH2-5 [1], revealing a four helical bundle (with no information for TMH1). Later, zero-length cross-linking experiments by the Fillingame group [2] identified contacting residues from all pairs of helical combinations of TM2-TM5 (6), supporting the earlier 4 helical bundle topology. 7 of the 8 cross-linked pairs from Schwem et al., are either exactly the same pair (L120-I246) or adjacent to many pairs in the top L intra a-subunit evolutionary couplings (ECs)*, see Figure 7. (Adjacent is defined as neighboring residues (±3) on chain or one helical turn.) * of ∼ L^2^ total, see previous work and work of others for why this number is chosen.

Information from the cross-linking was not used to construct the model and no presumed membrane topology was used as input. The consistency we report is based on the correspondence between the helical-helical contacts from the experiments and evolutionary couplings together with the resulting contacts in the computed model. These can be examined in detail in the supplementary material. We have clarified this part of the analysis and provided more detail.

Reviewer 2:

[…] I feel the paper could be substantially improved if the following points were addressed:

*1) Please support the statement "Benchmark calculations here ... indicate that the number of sequences in the alignment is critical..." with a Figure showing overall performance versus # of sequences (*Figure 3
*shows dependency of accuracy on relative rank for only two examples)*.

*2) Show a main Figure comparing predicted contacts against residue distances in the 30 known crystal structures of complexes that had a sufficient number of sequences (*Figure 2
*only shows contact plots for two examples, and structural pictures for several more, but no overall quantification). This type of analysis could also help to support the statement "For the top ranked benchmark complexes, the majority of the top 5 ECs is correct to within 8A..." with a more quantitative analysis (*Table 1
*only shows 7 of the 30 complexes in the dataset;*
[Supplementary-material SD3-data]
*shows all individual contact maps, but no overall quantification)*.

*3) "... these benchmarks show ... and demonstrate the criteria needed for successful prediction of unknown interactions". To support this statement, it would be useful to have a main Figure / Table for each important criterion, analyzed over an entire benchmark set (as for the number of sequences in point 1)*.

The claims in the paper are now supported by more explicit quantitative analysis, reported in the main figures, the text and supplementary material. Specifically we show a score that normalizes the ECs for the number of sequences and length of protein. This score allows a clear comparison of EC pairs across complexes so that one threshold can be applied across all complexes and precision can be measured for each complex and summarized overall. This material is provided in Figure 2 and Figure 2—figure supplement 1 and Figure 2—figure supplement 2 (all EVcomplex scores, plots of raw scores showing length and number of sequences dependence, plots of EVcomplex scores versus accuracy), Supplementary data and Materials and methods.

*4) Please add metrics to support statements that ECs are "completely consistent" with crosslinking data, or that a coupling "coincides with experimental evidence". The comparison Table in Supplementary Figure 6 could be presented in more quantitative terms in the main text (for example, what does "crosslinking neighborhood" mean?)*.

In summary, 77%/67% of the above threshold (13) predicted inter-protein contacts between subunit-a and subunit-b are in the neighborhood of residues identified in cross-linking studies, where neighborhood is defined as within 6/3 residues on the chain of the cross-linked residue ([Supplementary-material SD6-data]). The a239V- b16L cross-link is exactly one of the ECs. For 3/13 with no evidence, the proposed residue – residue contact between the subunits is consistent with the other cross-links: two are between midway on aTMH1 with midway of the membrane part of the b-subunit (aV50, aG53 with b13A), one is between aTMH3 and the N terminal end of the b-subunit (a155L, b7I) (these regions have not been tested to our knowledge). The 4^th^ predicted EC with no cross-link support is between a66K and b68a. This EC is most likely a false positive as residue 68 in the b-subunit may be too far from the membrane surface to be plausible. Taking the body of work together from the experimental side [3-8], there are a number of specific regions of the a-subunit that are in contact with one or both of the b-monomers, and all of these regions are represented in the top evolutionary couplings; namely the first cytoplasmic loop, aTMH2, upper portion, aTMH3 and the periplasmic end of aTMH5, all of which are represented in the EVcomplex pairs ([Supplementary-material SD6-data]).

These clarifications are reflected in the main text and [Supplementary-material SD6-data].

See reply to Reviewer 1 and Figure 7 for more detail on the ECs and crosslink studies within the a-subunit.

5) What are the possible reasons for false-positive co-variation that does not correspond to contact?

There are two categories of false positives:

1) There is co-evolution between residues that are not close in a 3D structure.

i) This could be functional coupling – e.g. cases of information transmission through membrane proteins, allosteric signaling and indeed we have seen this anecdotally in a number of very interesting cases. The challenge is to distinguish this from other false positives.

ii) The current crystal structures do not capture all the conformations of the protein(s). This is very clear in examples such as alternating-access membrane transporters that alter their conformations substantially

2) The assumptions used for the input sequence alignments are wrong

i) The interactions in the alignments are not homologous despite sequence similarity of the component parts.

ii) The sequence diversity is insufficient to capture realistic parameters in the global model.

Deconvolution of these different causes of false positives is a critical part of future work. We have made these points clearer in the Results and the Discussion.

*6) The manuscript deals entirely with pairwise co-evolution, and seems to neglect the possibility of higher-order complexity in the sequence pattern*.

There could well be higher order correlations that the current method is unable to determine at this stage. The number of sequences needed for a global model such as this to determine accurate higher order covariation would be at least an order of magnitude higher than the requirement for the current method. Indeed there are insufficient sequences for many of the currently known *E. coli* interactions (Discussion) and see response to Reviewer 1 – comment 3. It will be interesting to see what the extent of this higher order dependence might be when enough sequences become available.

Reviewer 3:

*The manuscript presents an upbeat and optimistic view of the success of the method, with a relatively small amount of critical discussion and little if any investigation of what one can learn from cases in which it does not perform as expected*.

We have updated the manuscript to include far more quantitative analysis, see replies to Reviewer 2 comments 1-4 above. We have also balanced the enthusiasm with a more specific description of the current limitations, that include sequence availability but also some that require algorithmic development (see Discussion). Critical discussion on the classification of false positives is now more clearly organized in the manuscript; see response to Reviewer 2, comment 5.

*(For example, a list of possible reasons for false-positive co-variation that does not correspond to contact is given, but an analysis of actual instances is not.) Likewise, a set of methods is presented that is relatively particular, and the reason why these specific methods were chosen over other possibilities is not provided*.

There is a large body of work that has implemented other methods, both force field–based docking and other sequence based methods to explore co-variation e.g. mutual information and phylogeny to detect co-variation of residues across proteins. We have now added more discussion and references to this oeuvre in the Introduction, including new references to the body of work by the Valencia group that used correlations in multiple sequence alignments [9-10], and those suggested by Reviewer 1. Many other methods have addressed somewhat different issues, for instance, identification of interacting proteins (not residue level information), identification of residue patches rather than specific interactions, concentration on docking with hybrid approaches for known interactions or were relatively less successful (e.g. older work). For this reason it is hard to make direct comparisons. The unique aspect to this work is that it can resolve the question of interacting partners together with residue level information if the sequence diversity and space is sufficient. The EVcomplex approach, along with that of the recent work by the Baker group [11] also does this from sequences alone. Since experimentally determined structures may be a somewhat biased set of all protein interactions, our method has the advantage of not learning on known complex structures.

We look forward to the combination of the co-evolution approach together with existing computational and experimental approaches to accelerate the field.

*Moreover, the manuscript deals entirely with pairwise co-evolution, and seems to neglect the possibility of higher-order complexity in the sequence pattern. The manuscript could be substantially strengthened by adding analyses and insights in these issues*.

See response to Reviewer 2, comment 6.

References

1 Rastogi, V. K. & Girvin, M. E. Structural changes linked to proton translocation by subunit c of the ATP synthase. *Nature*
**402**, 263-268, doi:10.1038/46224 (1999).

2 Schwem, B. E. & Fillingame, R. H. Cross-linking between helices within subunit a of Escherichia coli ATP synthase defines the transmembrane packing of a four-helix bundle. *The Journal of biological chemistry*
**281**, 37861-37867, doi:10.1074/jbc.M607453200 (2006).

3 Long, J. C., DeLeon-Rangel, J. & Vik, S. B. Characterization of the first cytoplasmic loop of subunit a of the Escherichia coli ATP synthase by surface labeling, cross-linking, and mutagenesis. *J Biol Chem*
**277**, 27288-27293, doi:10.1074/jbc.M202118200 (2002).

4 Fillingame, R. H. & Steed, P. R. Half channels mediating H transport and the mechanism of gating in the F sector of Escherichia coli FF ATP synthase. *Biochimica et biophysica acta*, doi:10.1016/j.bbabio.2014.03.005 (2014).

5 DeLeon-Rangel, J., Ishmukhametov, R. R., Jiang, W., Fillingame, R. H. & Vik, S. B. Interactions between subunits a and b in the rotary ATP synthase as determined by cross-linking. *FEBS Lett*
**587**, 892-897, doi:10.1016/j.febslet.2013.02.012 (2013).

6 Fillingame, R. H., Angevine, C. M. & Dmitriev, O. Y. Mechanics of coupling proton movements to c-ring rotation in ATP synthase. *FEBS Lett*
**555**, 29-34 (2003).

7 Brandt, K. *et al.* Individual interactions of the b subunits within the stator of the Escherichia coli ATP synthase. *The Journal of biological chemistry*
**288**, 24465-24479, doi:10.1074/jbc.M113.465633 (2013).

8 McLachlin, D. T. & Dunn, S. D. Disulfide linkage of the b and delta subunits does not affect the function of the Escherichia coli ATP synthase. *Biochemistry*
**39**, 3486-3490 (2000).

9 Pazos, F., Helmer-Citterich, M., Ausiello, G. & Valencia, A. Correlated mutations contain information about protein-protein interaction. *Journal of molecular biology*
**271**, 511-523, doi:10.1006/jmbi.1997.1198 (1997).

10 Pazos, F. & Valencia, A. In silico two-hybrid system for the selection of physically interacting protein pairs. *Proteins*
**47**, 219-227 (2002).

11 Ovchinnikov, S., Kamisetty, H. & Baker, D. Robust and accurate prediction of residue-residue interactions across protein interfaces using evolutionary information. *eLife*
**3**, e02030, doi:10.7554/eLife.02030 (2014).